



# Analysis and calibration of optimal power balance rotor-effective wind speed estimation schemes for large-scale wind turbines

Atindriyo Kusumo Pamososuryo[1], Fabio Spagnolo[2], and Sebastiaan Paul Mulders[1]

[1]Delft Center for Systems and Control, Delft University of Technology, Mekelweg 2, 2628 CD Delft, The Netherlands
[2]Vestas Wind Systems A/S, Hedeager 42, 8200 Aarhus N, Denmark

**Correspondence:** Atindriyo Kusumo Pamososuryo (A.K.Pamososuryo@tudelft.nl)

**Abstract.** Modern wind turbines' size growth creates challenges in their control system design, particularly due to greater wind variability across larger rotor areas. As modern turbine control systems rely on the availability of accurate wind speed information, the increasing unrepresentativeness of pointwise measurement devices, such as anemometers, necessitates the incorporation of more representative rotor-effective wind speed (REWS) estimation. Classical REWS estimators, based on static

power relations, often fail to account for dynamic changes, leading to inaccurate estimation. To overcome these challenges, this paper introduces a power balance-based REWS estimation framework and splits the estimation problem into two modules: an *aerodynamic power estimator* and a *wind speed estimate solver*. Two possible aerodynamic power estimation techniques are discussed based on *numerical derivative* and *state estimation*. As state estimator calibration remained a challenge for varying wind turbine sizes, a gain-tailoring method for the performance calibration throughout a range of modern wind turbine sizes

has been derived for the state estimation-based aerodynamic power estimator. Two types of wind speed estimate solvers are analyzed, namely the *continuous* and *iterative single-step* methods. From the two modules, the best-performing methods—the state-estimation aerodynamic power estimator and iterative single-step wind speed solver—are chosen to form the optimal power balance REWS estimator. The combined optimal estimator is validated through OpenFAST simulations of the NREL-5MW and IEA-22MW turbines and compared against a baseline method. The proposed method demonstrates good tracking of

the REWS, better noise resilience, and convenient estimator gain calibration across different turbine sizes.

## 1 Introduction

With the increasing demand for clean and renewable wind energy for the provision of electricity worldwide, there is a trend to upscale wind turbine sizes (Global Wind Energy Council, 2024). Greater wind turbine rotor swept areas enable more wind energy to be harnessable, resulting in increasing power production per unit turbine and effectively lowering the so-called

levelized cost of energy—thus making wind turbines more competitive in the energy market (Burton et al., 2011; Veers et al., 2019).

Regardless of the potential economic benefit, the task of controlling wind turbines with larger rotors is becoming more of a challenge, especially when accurate information of wind speed is crucial to ensure high controller performance, e.g., for gain-scheduling (Kumar and Stol, 2009; Koerber and King, 2013), feedforward control (Van Engelen and Van der Hooft, 2003;



Koerber and King, 2013), or tip-speed ratio tracking (Bossanyi, 2000; Ortega et al., 2013; Brandetti et al., 2022), to mention a few. This is mainly caused by the greater spatial variability across the rotor-swept area for larger turbines. Thus, pointwise wind speed information, such as provided by anemometers downstream at the nacelle, becomes increasingly unrepresentative, not to mention the presence of highly perturbed wind flow by the rotating turbine blades (Soltani et al., 2013). On the other hand, being the main driving force of a wind turbine, the deduction of more representative wind speed information via the turbine dynamics

has been seen as a viable alternative (Boukhezzar and Siguerdidjane, 2011). To be more exact, other available measurements, namely rotor speed, generator torque signal, and blade pitch position, can be made use of to provide the so-called rotor-effective wind speed (REWS) estimate (Østergaard et al., 2007).

    Early REWS estimation studies (see Østergaard et al. (2007); Soltani et al. (2013) and references therein), for a large part, utilize the static relation between the produced power and the REWS by omitting the always-occurring dynamical changes in

the rotor speed. Resultingly, these REWS estimators cannot provide accurate estimations of the aerodynamic torque in transient conditions due to the neglected dynamic information.

    To address the aforementioned shortcoming, later REWS estimation studies, incorporating the rotor acceleration information in their framework, arose in the literature. In Bossanyi (2000), the REWS estimate is obtained by firstly estimating the aerodynamic torque by reformulation of torque balance drivetrain dynamics, which account for rotor acceleration and drive-

train inertial information. Then, given a priori knowledge of the aerodynamic torque coefficient table, it is possible to deduce the information on the wind speed[1]. The work of Van Engelen and Van der Hooft (2003) and Boukhezzar and Siguerdidjane (2011) adopt a similar estimation approach, where for the wind speed estimate solver, the Newton-Raphson algorithm, being an *iterative single-(time)step* method, is utilized. In Ortega et al. (2013), an Immersion and Invariance method for the wind speed estimation is employed and its global convergence guarantee is provided. This method seeks to nullify the error between

measured rotor speed and its estimate, the latter being the integrated difference between (inertia-scaled) generator torque and aerodynamic torque estimate, by a proportional compensator (plus an integrator, as extended by Liu et al. (2022)). Having canceled the rotor speed estimation error, the wind speed estimate is then obtained in a *continuous* manner. The continuous method, in comparison with the iterative method, is considered to be a multiple time-step solving scheme.

    Nevertheless, stability analysis of the continuous solver following a discretization has remained unaddressed, to the best of

the authors' knowledge. Moreover, the performance comparison between the two wind speed estimation solving methods has received little attention in the literature.

    With regard to the aerodynamic torque or power estimator, the work of Østergaard et al. (2007) is of particular interest. Two ways to obtain the aerodynamic torque estimate are studied therein, namely by (filtered) *numerical derivative* or by *state estimation*. The former is associated with the numerical differentiating method used to obtain rotor acceleration estimate

from measured rotor speed. This provides a necessary 'ingredient' to reconstruct aerodynamic torque given a priori inertia information and generator torque input. The latter method provides two cascaded observers. The inner loop estimates the

---

[1]In Bossanyi (2000), the information on the tip-speed ratio corresponding to the estimated aerodynamic torque is the main estimation output. Thus, given rotor speed measurements, the wind speed estimate can be straightforwardly derived from this tip-speed ratio estimate (as explained in Østergaard et al. (2007)).



unmeasurable turbine states by Kalman filtering, while the outer one estimates the aerodynamic torque by a proportional-integral compensator structure. Although performance comparisons of both methods are provided, little attention was paid to the effect of noisy measurements on the aerodynamic torque estimators, which might deteriorate the ensuing wind speed estimation. To facilitate calibration of wind speed estimation by state estimation, the work of Moustakis et al. (2019) proposed a machine-learning-based Bayesian optimization approach. Nonetheless, since it remained unclear how to properly tune such a wind speed estimator, a Bayesian optimization approach, which is a global optimization machine learning algorithm, was adopted in the study.

However, optimal wind speed estimator tuning for a single turbine might not necessarily translate into optimal performance when applied to a different turbine. Given the accelerated growth in modern wind turbine sizes, there is a need to derive a calibration methodology to ensure optimal estimator tuning throughout these turbines.

Furthermore, based on the above literature review, four possible combinations of aerodynamic torque (or aerodynamic power, as made clearer shortly) estimator and wind speed estimate solver can be constructed with the optimal combination left undetermined. Moreover, validation in realistic simulation settings of such an optimal wind speed estimator combination—calibrated for wind turbines of various sizes—needs to be performed. It is also worth noting that the aforementioned works on various wind speed estimator schemes are based on the torque balance modeling of the wind turbine drivetrain. While utilizing such torque-based coordinates has been widely used for wind turbine control designs in the literature, employing power-based terms to represent the wind turbine dynamics is commonly used within the industry (Hovgaard et al., 2015; Odgaard et al., 2017; Brandetti et al., 2022; Mulders et al., 2023a; Pamososuryo et al., 2023). The current work thus provides a wind speed estimator framework, for a major part, in terms of power balance dynamics. That said, adopting this work into the torque balance framework is straightforward. The following outlines the contributions of this work

1. Provide a thorough analysis for numerical derivative- and state estimation-based aerodynamic power estimators given noisy measurement settings;

2. Formalizing a calibration methodology for state estimation-based aerodynamic power estimator for a range of modern wind turbine sizes;

3. Deriving iterative and continuous wind speed estimate solvers, while providing frequency-domain stability analysis for the latter mentioned method;

4. Identifying the optimal wind speed estimator structure out of the proposed aerodynamic power estimators and wind speed estimate solvers;

5. Provide a mid-fidelity validation of the selected optimal estimator under realistic conditions for multiple wind turbine sizes.

The remainder of this paper is structured as follows: In Section 2, preliminaries required for this paper, being the notation convention, key reference wind turbine properties, and assumptions used throughout the paper, are explained. Section 3 touches



**Table 1.** Reference wind turbines' key physical properties. Those of the NREL-5MW are taken from Jonkman et al. (2009), the IEA-10MW from Bortolotti et al. (2019), the IEA-15MW from Gaertner et al. (2020), and the 22MW turbine from Zahle et al. (2024).

| Turbine parameter | Reference wind turbine | | | |
| --- | --- | --- | --- | --- |
| | NREL-5MW | IEA-10MW | IEA-15MW | IEA-22MW |
| Rated power, $P_{\mathrm{g,rated}}$ (MW) | 5 | 10 | 15 | 22 |
| Rotor diameter, $D$ (m) | 126 | 198 | 240 | 280 |
| LSS-equivalent inertia, $J$ (kg m$^2$) | 43,702,538 | 160,342,052 | 312,456,272 | 752,272,514.5 |
| Gearbox ratio, $G$ (−) | 97 | 1 | 1 | 1 |

upon the closed-loop wind turbine model and the proposed power balance REWS estimation framework. Section 4 and 5 covers
several potential options for the aerodynamic power estimator and wind speed estimate solver subcomponents, respectively,
where thorough analyses and low-fidelity numerical demonstrations are given. In Section 6, the proposed combination of the
aerodynamic power estimator and wind speed estimate solver subcomponents are validated using higher-fidelity wind turbine
simulation results. Finally, the conclusions and recommendations of this work are laid out in Section 7.

## 2  Preliminaries

### 2.1  Notations

In this section, frequently used notations in this paper are defined. Time dependency in the continuous domain is indicated by
the time variable $t$ and in the discrete-time domain by the time-step variable $k$. Quantities in the continuous frequency/Laplace
domain are indicated by $s$ notation and those in the discrete $z$-domain with $z$. A signal's first time-derivative is denoted by $(\dot{\bullet})$,
the $(\hat{\bullet})$ notation indicates an estimated quantity, $(\bar{\bullet})$ indicates a quantity at its steady-state, and $(\tilde{\bullet})$ denotes a signal corrupted
by noise. Constants associated with the optimal power coefficient, design tip-speed ratio, and fine pitch angle are indicated by
$(\bullet^\star)$.

### 2.2  Key Reference Wind Turbine Properties

As mentioned earlier in Section 1, wind speed estimator calibration for real-world industrial turbines is presented in this work.
Therefore, a wide range of wind turbine power capacities, which are at the present day represented by the available reference
wind turbine models ranging from 5 to 22MW, is considered to showcase the applicability of the current study on a wide range
of relevant-sized wind turbines. For that purpose, several reference wind turbines are considered in this study and their key
physical properties are summarized in Table 1.

For later convenience, empirical relations have been derived between the turbines' rotor diameters $D$, power ratings $P_{\mathrm{g,rated}}$,
as well as their low-speed-shaft equivalent inertias $J$. By drawing such relations, it is possible to account for more turbine
dimensions, power ratings, and inertias other than that of the reference turbines. To this end, the key properties of the reference



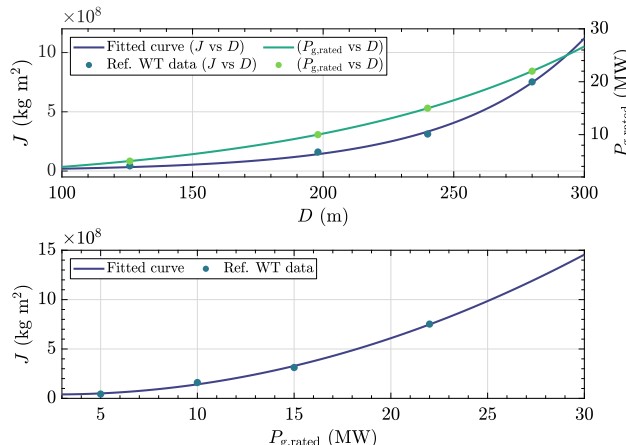

**Figure 1.** Curve fitting results. Reference wind turbine datapoints are depicted by the dots and the fitted curves are indicated by the lines. The top plot is the mapping from the rotor diameter to the inertia (left y-axis) and rated generator power (right y-axis). The bottom plot is the mapping from the rated generator power to the inertia.

wind turbines in Table 1 are made use of to obtain the following fitted functions

$$J(D) = \left(2.581 \cdot 10^6 \cdot e^{0.02024D}\right) \quad \text{kg m}^2, \tag{1a}$$

$$P_{\text{g,rated}}(D) = \left(1.491 \cdot 10^6 \cdot e^{0.009613D}\right) \quad \text{W}, \tag{1b}$$

$$J(P_{\text{g,rated}}) = \left(9.979 \cdot 10^7 P_{\text{g,rated}}^2 + 2.843 \cdot 10^8 P_{\text{g,rated}}\right.$$

$$\left. + 2.424 \cdot 10^8\right) \quad \text{kg m}^2. \tag{1c}$$

For the above fits, the coefficient of determination $\mathcal{R}^2 > 0.99$ is ensured. As the reference wind turbines represent industrial turbine designs, it is expected that the derived fits hold acceptable prediction quality for industrial turbines.

### 2.3 Assumptions

**Assumption 1.** *There are no electromechanical losses in the drivetrain; that is, the generator and gearbox efficiencies are assumed to be* $100\%$.

**Assumption 2.** *The wind turbines analyzed in this study operate solely in the partial-load (below-rated) region, where the generator power is controlled. The blade pitch angle is set to its constant fine pitch position. Consequently, the power coefficient depends solely on the tip-speed ratio.*

**Assumption 3.** *The power coefficient represents the exact steady-state aerodynamic characteristics of the actual rotor.*

**Assumption 4.** *In the low-fidelity simulations provided throughout this study, the power coefficient of all the considered reference wind turbines (see Table 1) is equal to that of the NREL-5MW reference wind turbine so as to enable a clear analysis and comparison of the results between the various considered turbines.*



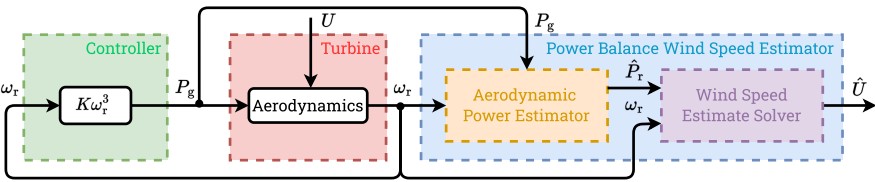

**Figure 2.** The general scheme of the power balance wind speed estimation considered in this study. The wind turbine (red block) is operated in a closed loop with a $K\omega_\mathrm{r}^3$ controller (green block), whereas the power balance wind speed estimator (blue block) is in an open-loop configuration with the turbine. The power balance wind speed estimator is subdivided into the aerodynamic power estimator (yellow block) and wind speed estimate solver (purple block).

**Assumption 5.** *The drivetrain inertia value at the low-speed shaft side is assumed to be an a priori known parameter.*

## 3 Closed-Loop Wind Turbine Model and Rotor-Effective Wind Speed Estimation Framework

Figure 2 presents the overall scheme considered in this work, in which the wind turbine is controlled by a partial-load controller along with a power balance wind speed estimator. The wind speed estimator, being the main focus of the analysis in this work, is connected in an open loop to the closed-loop system. The red block represents the wind turbine, the green block contains the controller, and the blue block is the power balance wind speed estimator considered in this study.

Sections 3.1 and 3.2 provide the required theory used in this paper by outlining the first two subsystems, followed by defining
the wind speed estimator. Then Section 3.3 addresses the decomposition of the estimator into several subcomponents, providing a framework for the remainder of the work presented in this paper.

### 3.1 Single-Degree-of-Freedom Wind Turbine Model and Optimal Controller

In this work, single-degree-of-freedom power-balance drivetrain dynamics are considered as a simplified representation of a wind turbine as follows

$$J\omega_\mathrm{r}(t)\dot{\omega}_\mathrm{r}(t) = P_\mathrm{r}(t) - P_\mathrm{g}(t), \tag{2}$$

where $J \in \mathbb{R}$ is the low-speed shaft (LSS) equivalent inertia, $\omega_\mathrm{r} \in \mathbb{R}$ the rotor angular speed, and $P_\mathrm{g} \in \mathbb{R}$ the generated power. The aerodynamic power is given by the nonlinear relation

$$P_\mathrm{r}(t) = \frac{1}{2}\rho A_\mathrm{r} C_\mathrm{p}(\lambda(t)) U(t)^3, \tag{3}$$

in which $\rho \in \mathbb{R}$, denotes the air density, $A_\mathrm{r} \in \mathbb{R}$ the rotor area, and $U \in \mathbb{R}$ the REWS (Soltani et al., 2013). The power coefficient
at the fine pitch $C_\mathrm{p} : \mathbb{R} \to \mathbb{R}$ (see Assumption 2) is a nonlinear mapping from the non-dimensional tip-speed ratio (TSR), defined as

$$\lambda(t) = \frac{\omega_\mathrm{r}(t)R}{U(t)}, \tag{4}$$



with $R \in \mathbb{R}$ as the rotor radius.

The drivetrain system outputs $\omega_\mathrm{r}$, which is then fed into the optimal torque controller (Bossanyi, 2000), often known as the '$K\omega_\mathrm{r}^2$' controller. However, partial-load controller design is not the main focus of this study; hence, the $K\omega_\mathrm{r}^2$ controller is deemed sufficient for the goal of this work. That said, this work is equally applicable to more advanced partial-load controllers available in the literature, such as tip-speed ratio tracking schemes, e.g., Brandetti et al. (2023) and Lazzerini et al.. The $K\omega_\mathrm{r}^2$ controller, in its generator-power equivalence, is expressed as

$$P_\mathrm{g}(t) = K\omega_\mathrm{r}^3, \tag{5}$$

where

$$K = \frac{\pi \rho R^5 C_\mathrm{p}^\star}{2\lambda^{\star 3}},$$

is the optimal control gain. The notation $\lambda^\star$ indicates the design TSR, corresponding to the optimal power coefficient $C_\mathrm{p}^\star := C_\mathrm{p}(\lambda^\star)$. Based on the expression (5), in the remainder of this paper, as well as in Fig. 2, this optimal controller is referred to as the '$K\omega_\mathrm{r}^3$'.

## 3.2 Power Balance Wind Speed Estimation General Concept

This section establishes the REWS estimation framework that forms a basis for the remainder of this paper. The rationale behind the power balance REWS estimator presented herein lies in the retrievability of the wind speed information by asymptotic minimization of an error term between the aerodynamic power and its estimate, in which Assumption 3 holds, that is

$$U(t) = \lim_{t \to \infty} \underset{\hat{U}(t)}{\arg\min} |e_\mathrm{p}(t)|, \tag{6}$$

where $\hat{U} \in \mathbb{R}$ denotes the REWS estimate. The notation $e_\mathrm{p} \in \mathbb{R}$ is the said estimation error, defined as

$$e_\mathrm{p}(t) = P_\mathrm{r}(t) - \frac{1}{2}\rho A_\mathrm{r} C_\mathrm{p}\left(\hat{\lambda}(t)\right)\hat{U}(t)^3, \tag{7}$$

where the second term on the right-hand side of the equation is the aerodynamic power estimate based on (3), utilizing $\hat{U}$ in place of $U$. The challenge of obtaining an estimate of $P_\mathrm{r}$ and solving the optimization problem of (6) is explained in further detail in the next section.

**Remark 1.** *Note that* (6) *will not be achieved in the presence of discrepancies between the used and actual $C_\mathrm{p}$ tables. Such disparities will lead to a biased wind speed estimate as reported in Brandetti et al. (2022). Readers interested in the details of such ill-conditioning are thus referred to the study.*

## 3.3 Wind Speed Estimator Subcomponent Partitioning

Now that the closed-loop controlled wind turbine and wind speed estimation problem has been defined, the power balance REWS estimator explained in Section 3.2 can be partitioned into two subcomponents to allow for both rigorous analysis and





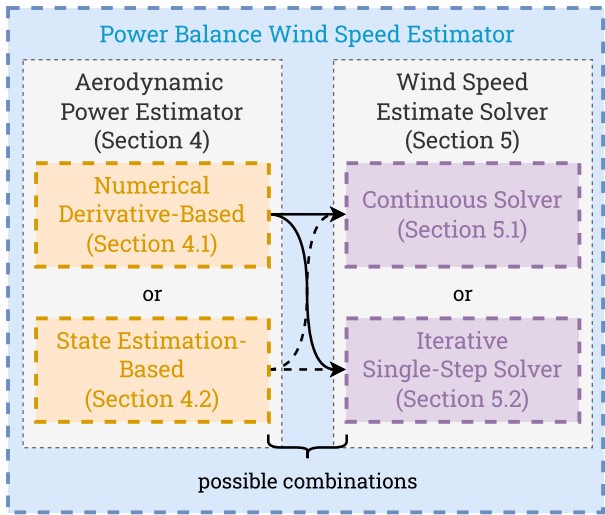

**Figure 3.** Power balance wind speed estimation partitioning. The left column contains the potential aerodynamic power estimators to be connected with the potential wind speed estimate solvers within the right column, thereby creating four possibilities for combining the two subcomponents.

effective estimation schemes: (i) aerodynamic power estimator and (ii) wind speed estimate solver. Figures 2 and 3 depict this partitioning, the latter of which details the possible techniques to realize these two subcomponents. Similar separation approaches have also been adopted in the literature, e.g., in Van Engelen and Van der Hooft (2003) and Østergaard et al. (2007). The current study provides a more in-depth analysis of the subcomponents.

The first block provides an estimate of the aerodynamic power, which—in contrast to the measured generator power— is more challenging to obtain, as will be explained shortly. Fortunately, such information can still be obtained based on the available measurements and is the concern of the orange blocks in Figs. 2-3. By rearranging (2) and replacing specific variables with their estimated representations, one obtains

$$\hat{P}_{\mathrm{r}}(t) = J\omega_{\mathrm{r}}(t)\hat{\dot{\omega}}_{\mathrm{r}}(t) + P_{\mathrm{g}}(t)\,. \tag{8}$$

Note that $\hat{P}_{\mathrm{r}}$ and $P_{\mathrm{g}}$ are equal to each other in steady state; however, due to the variable nature of the wind and rotor speed, omitting the rotor acceleration term entirely from this calculation means losing valuable dynamic information (Østergaard et al., 2007; Soltani et al., 2013). Therefore, taking into account the dynamics by the utilization of the rotor acceleration and the $J\omega_{\mathrm{r}}\dot{\omega}_{\mathrm{r}}$ terms enhances the accuracy of REWS estimate during both steady-state and transient conditions.

In practice, however, the rotor acceleration is not directly measurable, not to mention that it is challenging to obtain a good

estimate of this quantity, $\hat{\dot{\omega}}_{\mathrm{r}}$, due to the noisy nature of measured signals in practice. To retrieve $\hat{\dot{\omega}}_{\mathrm{r}}$, one may resort to *numerical derivative* of $\omega_{\mathrm{r}}$ or *state estimation* methods as depicted in Fig. 3. Subsequently, the aerodynamic power estimate $\hat{P}_{\mathrm{r}}$ is obtained by solving (8).





**Remark 2.** *At this point, two aerodynamic power estimate terms have been introduced. One being $\hat{P}_\mathrm{r}$, defined in* (8) *and another being the second term on the right-hand side of* (7). *To prevent any confusion, the term 'aerodynamic power estimate' is used to refer to the former, whereas that of the latter from hereon is referred to as the '$\hat{U}$-dependent aerodynamic power estimate.'*

Independent of how $\hat{P}_\mathrm{r}$ is retrieved, such information, together with $\omega_\mathrm{r}$ measurements, is then fed into the wind speed estimate solver subcomponent indicated as the purple blocks in Fig. 3. Solving for the estimated wind speed is achieved in two ways in this work: *continuous* (e.g., as used in Ortega et al. (2013) and Liu et al. (2022)) or *iterative single-step* manner (e.g., by Newton-Raphson methods as done in Van Engelen and Van der Hooft (2003) and Boukhezzar and Siguerdidjane (2011)). A linear analysis of the continuous wind speed estimate solver in continuous time will be provided, in which the stability properties of the linearized solver dynamics are derived. Furthermore, the effects of various discretization methods on the system, especially on the mentioned stability properties, are evaluated. The single-step approach solves the wind speed estimate in a similar way to the former without the need for high solver gains, potentially causing stability issues to obtain good estimation quality without phase lags. Its wind speed estimation quality is determined by the choice of error tolerance parameters and iteration budget.

The aforementioned options for each of the subcomponents, therefore, allow for several possible combinations in which the power balance wind speed estimator can be constructed, as illustrated in Fig. 3; however, the optimal combination is yet to be found. To that end, in the respective Sections 4 and 5, the derivations of the two aerodynamic power estimators and the wind speed estimate solvers are provided, where also their performance is evaluated.

## 4 Aerodynamic Power Estimator

As discussed previously, reconstructing aerodynamic power from available measurements is an essential step in obtaining an accurate REWS estimate in both steady-state and dynamic transient conditions. To this end, the most challenging part is obtaining an accurate estimate of the rotor acceleration $\hat{\dot{\omega}}_\mathrm{r}$, which is the main concern of this section. Two approaches are considered herein; first, the numerical derivative-based method is examined in Section 4.1. Later in Section 4.2, the state estimation-based technique is discussed. In Section 4.3, the numerical comparisons for both methods are evaluated.

### 4.1 Numerical Derivative-Based Technique

To obtain an estimate of the rotor acceleration, a numerical derivative is applied to the measured $\omega_\mathrm{r}$. In the frequency domain, this is represented as follows

$$\hat{\dot{\Omega}}_\mathrm{r}(s) = F_\mathrm{nd,c}(s,\tau)\Omega_\mathrm{r}(s)\,, \tag{9}$$

in which $\Omega_\mathrm{r}$ and $\hat{\dot{\Omega}}_\mathrm{r}$, with a slight abuse of notation for the latter, are the respective Laplace-transformed variables of $\omega_\mathrm{r}$ and $\hat{\dot{\omega}}_\mathrm{r}$. The transfer function

$$F_\mathrm{nd,c}(s,\tau) = \frac{s}{\tau s + 1}\,, \tag{10}$$





in (9) is the filtered derivative[2] in accordance with IEEE 421.5-2016 standard (IEEE, 2016), with a unity derivative gain. The parameter $\tau \in \mathbb{R}^{\geq 0}$ is the time constant of the numerical derivative.

In its implementation, the numerical derivative (10) is discretized via the Backward Difference method. Thus, the discrete-time transfer function of the filter is

$$F_{\mathrm{nd,d}}(z,\tau) = \left(\frac{1}{\tau}\right)\frac{1 - z^{-1}}{1 + h/\tau - z^{-1}}\,, \tag{11}$$

where $h$ denotes the sampling time.

As $\tau$ is the only tuning parameter for the filter (11) it plays a crucial role. For instance, setting $\tau = 0$ casts $F_{\mathrm{nd,d}}$ into a pure differentiator. This enables infinite amplification at the high-frequencies (including noise), that is propagated to $\hat{\dot{\omega}}_{\mathrm{r}}$, which is undesired. Having too large $\tau$ is also unwanted as $\hat{\dot{\omega}}_{\mathrm{r}}$ and the subsequent $\hat{P}_{\mathrm{r}}$ may become less accurate despite the better noise resilience. Calibration of $\tau$ is, therefore, a trade-off between having an accurate rotor acceleration estimate and good noise suppression. In the following section, a numerical demonstration of such a trade-off is performed and analyzed.

### 4.1.1 Time Constant Selection: Accuracy and Noise Propagation

As stated above, the choice of $\tau$ may be helpful in suppressing the effects of noisy measurements often encountered in real-world scenarios. To provide a clearer picture on this aspect, $F_{\mathrm{nd,d}}$ is applied to the discrete-time rendition of (9), resulting in the filtered and differentiated rotor speed

$$\tilde{\dot{\hat{\omega}}}_{\mathrm{r}}(k) = \frac{\tilde{\omega}_{\mathrm{r}}(k) - \tilde{\omega}_{\mathrm{r}}(k-1) + \tau\tilde{\dot{\hat{\omega}}}_{\mathrm{r}}(k-1)}{\tau + h}\,, \tag{12}$$

with

$$\tilde{\omega}_{\mathrm{r}}(k) = \omega_{\mathrm{r}}(k) + v_{\omega_{\mathrm{r}}}(k)\,. \tag{13}$$

as the noisy rotor speed signal, where $v_{\omega_{\mathrm{r}}} \sim (0, \sigma^2_{v_{\omega_{\mathrm{r}}}})$ is an additive, Gaussian white noise with zero mean and variance $\sigma^2_{v_{\omega_{\mathrm{r}}}}$.

The impact of the noise propagated from $\tilde{\omega}_{\mathrm{r}}$ will affect the aerodynamic power estimate, as made evident in the following relation, obtained by substituting (12)-(13) into (8)

$$\hat{P}_{\mathrm{r}}(k) = J\tilde{\omega}_{\mathrm{r}}(k)\tilde{\dot{\hat{\omega}}}_{\mathrm{r}}(k) + P_{\mathrm{g}}(k)\,. \tag{14}$$

Note that the equation contains a multiplication between $\tilde{\omega}_{\mathrm{r}}$ and $\tilde{\dot{\hat{\omega}}}_{\mathrm{r}}$. This implies that the noise the former contains is multiplied with the one it propagated to the latter from the previous time step. This introduces a biased $\hat{P}_{\mathrm{r}}$ that varies on the noise variance because the product of a noise sequence with itself, although it is of zero mean, will give a nonzero mean[3]. A chosen $\tau$ may lessen the effects of such noise propagation but may deteriorate $\hat{P}_{\mathrm{r}}$ estimation performance and is a trade-off.

---

[2]That is, a pure differentiator combined with a first-order low-pass filter.

[3]Consider this product to be $\nu_{\omega_{\mathrm{r}}} = v_{\omega_{\mathrm{r}}}^{\top}v_{\omega_{\mathrm{r}}}$. The mean of $\nu_{\omega_{\mathrm{r}}}$ is thus equal to the variance of $v_{\omega_{\mathrm{r}}}$, namely $E[\nu_{\omega_{\mathrm{r}}}] = E[v_{\omega_{\mathrm{r}}}^{\top}v_{\omega_{\mathrm{r}}}] = \sigma^2_{v_{\omega_{\mathrm{r}}}}$, where $E$ is the expected value operator (Verhaegen and Verdult, 2007).



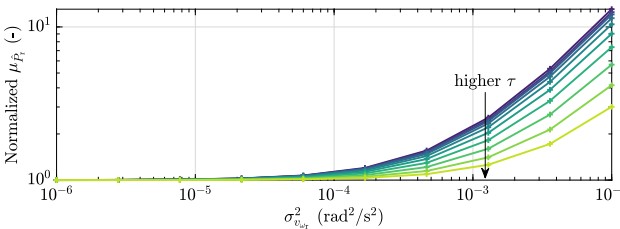

**Figure 4.** The mean of the aerodynamic power estimate $\mu_{\hat{P}_r}$, normalized with respect to that of a noiseless case. It is shown that with a high noise variance $\sigma^2_{v_{\omega_r}}$ the noise propagated by the numerical derivative from $\omega_r$ into $\hat{\omega}_r$ results in a high bias in $\hat{P}_r$. Nonetheless, the application of $\tau$ lessened the severity of the increased aerodynamic power estimation bias, shown by lower $\mu_{\hat{P}_r}$ as $\tau$ becomes higher.

To numerically demonstrate the effect of noisy measurements to (14), 400-second simulations, sampled at $h = 0.02$ s, were run for time constants and noise variances of $\tau \in [0, 10^{-1}]$ s and $\sigma^2_{v_{\omega_r}} \in [10^{-6}, 10^{-2}]$ rad$^2$/s$^2$, respectively. In addition, the NREL-5MW's inertia value is used (see Table 1), under the steady-state operating conditions $\bar{\omega}_r = 0.8$ rad/s and $\bar{P}_g = 1.647$ MW, the latter of which is computed using (5) with $C_p^\star = 0.469$ and $\lambda^\star = 6.53$. Figure 4 summarizes the statistical results of these simulations, where $\mu_{\hat{P}_r}$ is the mean of the aerodynamic power estimate.

It is apparent in the figure that greater noise variance leads to higher $\mu_{\hat{P}_r}$, representing added bias in the aerodynamic power estimate; nevertheless, employing high $\tau$ values alleviates such deterioration to some extent. Also implied in this observation is using $\tau = 0$ s (i.e., using a pure differentiator) is not desirable, especially in highly noisy environments, as this would lead to an infinite amplification of high-frequency components. In conclusion, attention needs to be paid to noisy $\omega_r$ conditions as the resulting biased $\hat{P}_r$ may undermine the REWS estimation in the end.

### 4.2 State Estimation-Based Technique

Besides the aforementioned numerical derivative technique to obtain $\hat{\omega}_r$ and, thus, $\hat{P}_r$, state estimation-based methods can also be employed. Obtaining $\hat{P}_r$ via state estimation can be proven to be more beneficial compared to the numerical derivative technique in the possessed freedom to trade-off sensitivity to noisy measurements with responsiveness by estimation gain tuning, however, might be more challenging in its implementation and calibration.

Despite the adopted power balance wind speed estimation framework, the state estimator employed in this section utilizes an internal model based on the torque balance variant of (2). Retaining the power variables in (2) would lead to the internal estimator dynamics being nonlinear such that it becomes necessary to obtain the system's Jacobians—adding complexities to the observer design. Therefore, to provide an aerodynamic power estimate, a reformulation is performed to obtain a torque-based estimator. To this end, the internal model is described as the following dynamics

$$\dot{\omega}_r(t) = \frac{T_r(t) - G T_g(t)}{J}, \tag{15}$$

where $G \in \mathbb{R}^+$ is the gearbox ratio of the drivetrain, $T_r = P_r/\omega_r$ is the aerodynamic torque, and $T_g = P_g/(\omega_r G)$ is the generator torque.



The dynamics (15) are then recast into the following discrete state-space form by employing Forward Euler discretization

$$x_s(k+1) = A_s x_s(k) + B_{s,u} u_s(k) + B_{s,d} d_s(k) + w_{\omega_r}(k),$$

$$y_s(k) = C_s x_s(k) + v_{\omega_r}(k), \tag{16}$$

with

$$x_s = \omega_r, \quad u_s = T_g, \quad d_s = T_r, \text{ and } \quad y_s = \omega_r,$$

as the respective estimator's state, input, disturbance, and output, and the state space matrices

$$A_s = 1, \quad B_{s,u} = -hJ^{-1}G, \quad B_{s,d} = hJ^{-1}, \text{ and } \quad C_s = 1,$$

where $\{x_s, u_s, d_s, y_s\} \in \mathbb{R}$ and $\{A_s, B_{s,u}, B_{s,d}, C_s\} \in \mathbb{R}$. Also included in (16) are the process noise $w_{\omega_r} \sim (0, \sigma^2_{w_{\omega_r}})$ with variance $\sigma^2_{w_{\omega_r}}$ and measurement noise $v_{\omega_r} \sim (0, \sigma^2_{v_{\omega_r}})$ with variance $\sigma^2_{v_{\omega_r}}$, both assumed to be uncorrelated, zero-mean, Gaussian white noise. The aerodynamic torque is considered to be an unknown input and, therefore, a subject of the estimation. Thus, it is recast as a *random-walk* process (Verhaegen and Verdult, 2007) as follows

$$T_r(k+1) = T_r(k) + w_{T_r}(k), \tag{17}$$

where $w_{T_r} \sim (0, \sigma^2_{w_{T_r}})$ is a zero-mean Gaussian white noise sequence with variance $\sigma^2_{w_{T_r}}$, uncorrelated to $w_{\omega_r}$ and $v_{\omega_r}$. One advantage of treating $T_r$ as a random-walk process is in the ease of design as no a priori information, such as aerodynamic torque coefficient, is needed. In particular, the Jacobian of this term is not necessary given that it is a nonlinear function of $v$ and $\omega_r$.

The general state space expression for the state estimator, augmenting (17) to (16) and also including feedback from the
measured system output $y = \omega_r$, is written as follows

$$\underbrace{\begin{bmatrix} \hat{x}_s(k+1) \\ \hat{T}_r(k+1) \end{bmatrix}}_{\hat{\boldsymbol{x}}_{s,\text{aug}}(k+1)} = \underbrace{\begin{bmatrix} A_s & B_{s,d} \\ 0 & 1 \end{bmatrix}}_{\boldsymbol{A}_{s,\text{aug}}} \underbrace{\begin{bmatrix} \hat{x}_s(k) \\ \hat{T}_r(k) \end{bmatrix}}_{\hat{\boldsymbol{x}}_{s,\text{aug}}(k)} + \underbrace{\begin{bmatrix} B_{s,u} \\ 0 \end{bmatrix}}_{\boldsymbol{B}_{s,\text{aug}}} T_g(k)$$

$$+ \underbrace{\begin{bmatrix} w_{\omega_r}(k) \\ w_{T_r}(k) \end{bmatrix}}_{\boldsymbol{w}_{s,\text{aug}}(k)} + \boldsymbol{L}(y(k) - \hat{y}_s(k)), \tag{18}$$

$$\hat{y}_s(k) = \underbrace{\begin{bmatrix} C_s & 0 \end{bmatrix}}_{\boldsymbol{C}_{s,\text{aug}}} \underbrace{\begin{bmatrix} \hat{x}_s(k) \\ \hat{T}_r(k) \end{bmatrix}}_{\hat{\boldsymbol{x}}_{s,\text{aug}}(k)} + v_{\omega_r}(k),$$

with $\boldsymbol{L} = [L_1, L_2]^\top$ as the observer gain vector. This gain can be determined either by a pole placement (Luenberger approach) or Kalman design, the latter of which is able to provide minimum-variance, unbiased state estimation by solving an algebraic



Riccati equation involving the noise covariance matrices. However, because the former provides more freedom to define the
state estimator's pole locations according to one's own optimal performance criterion, in this study, $\boldsymbol{L}$ is determined by pole
placement.

**Remark 3.** *Similar to the numerical derivative-based method, the state estimation scheme above can provide the aerody-
namic power estimate by making use of the relation $\hat{P}_\mathrm{r} = J\hat{\omega}_\mathrm{r}\dot{\hat{\omega}}_\mathrm{r} + P_\mathrm{g}$. Nevertheless, as in the state estimation-based scheme
considered in this section, the quantities $\hat{\omega}_\mathrm{r}$ and $\hat{T}_\mathrm{r}$ are accessible directly from the augmented state vector $\hat{\boldsymbol{x}}_\mathrm{s,aug}$ such that
the aerodynamic power estimate can be computed straightforwardly via $\hat{P}_\mathrm{r} = \hat{\omega}_\mathrm{r}\hat{T}_\mathrm{r}$. Thus, this approach is considered for the
remainder of this study.*

In the aforementioned Luenberger approach, $\boldsymbol{L}$ needs to be designed such that $\mathbf{A}_\mathrm{s,aug} - \boldsymbol{L}C_\mathrm{s,aug}$ has stable eigenvalues.
Such a condition guarantees $y - \hat{y}_\mathrm{s}$'s convergence to zero given observable $(\mathbf{A}_\mathrm{s,aug}, \boldsymbol{C}_\mathrm{s,aug})$. Evaluating this condition confirms
that it is satisfied for $hJ^{-1} \neq 0$, which is always the case in practical scenarios.

With regard to the estimator's performance, having a closer look into the characteristic polynomial of $\mathbf{A}_\mathrm{s,aug} - \boldsymbol{L}C_\mathrm{s,aug}$ may
shed some new insights, e.g., how the design can be applied for different wind turbine power ratings. Furthermore, it is also
somewhat known that a wind turbine's power rating is associated with its dimension and, thus, inertial properties (Rodriguez
et al., 2007), which, in this particular case, is the most influential as $h, G \ll J$. This indicates that a selected $\boldsymbol{L}$ suitable for a
wind turbine might give a different performance when applied to another turbine with a different power rating. Therefore, it is
crucial to find a gain-tailoring guideline in such a way that identical state estimator performance among different power ratings
or inertia values can be found.

The aforementioned considerations are addressed in the following sections: Section 4.2.1 provides the investigation into the
characteristic polynomial of the estimator. A numerical demonstration is presented in Section 4.2.2 to compare the performance
of the estimator with and without such a guideline.

### 4.2.1 State Estimator Characteristic Polynomial

This section covers the analysis of the characteristic polynomial for the state estimation-based aerodynamic power estimator
laid out in the previous section. Investigation into such a characteristic polynomial informs one about how, for instance, a
choice of estimator gain influences the estimator's natural frequency and damping. The characteristic polynomial is derived as
follows

$$\det(zI - \mathbf{A}_\mathrm{s,aug} + \boldsymbol{L}C_\mathrm{s,aug}) = z^2 + p_1 z + p_2 = 0, \tag{19}$$

in which the roots of the polynomial coefficients are parameterized as

$$p_1 = 2(h\omega_0\zeta_0 - 1) = L_1 - 2, \tag{20a}$$

$$p_2 = 1 - 2h\zeta_0\omega_0 + h^2\omega_0^2 = 1 - L_1 + \frac{h}{J}L_2, \tag{20b}$$





where $\omega_0$ and $\zeta_0$ are the respective natural frequency and damping ratio of the continuous-time characteristic equation[4], which by further manipulation of (20) leads to

$$\omega_0 = \sqrt{\frac{L_2}{hJ}}, \tag{21a}$$

$$\zeta_0 = \frac{L_1}{2h\omega_0}. \tag{21b}$$

It is directly evident from the above equations that to maintain constant $\omega_0$ and $\zeta_0$ for a range of different turbines, the ratio $L_2/J$ needs to be maintained constant under the assumption that $L_1$ and $h$ are equal for all turbines. Furthermore, it is more insightful to express $L_1$ and $L_2$ in terms of $\omega_0$ and $\zeta_0$ by rearranging (21) as follows

$$L_1 = 2h\zeta_0\omega_0, \tag{22a}$$

$$L_2 = hJ\omega_0^2. \tag{22b}$$

The relation above allows one to determine both gains based on specified $\omega_0$ and $\zeta_0$, but most importantly, it becomes clear that $L_2$ needs to be tailored based on the turbine inertia, especially if one desires to apply the estimator for different turbine sizes and power ratings as discussed in the previous section. In the following section, how this gain is tailored for a range of wind turbine inertias is discussed, and a numerical demonstration is also provided.

### 4.2.2 Constant and Tailored Estimator Gain Comparison

To numerically demonstrate the performance difference between constant and tailored $L_2$ over the considered range of turbines, 800-second simulations (sampled at $h = 0.02$ s) are performed with a turbulent wind with mean speed $U_{\mathrm{h}} = 7.5$ m/s and intensity $I_{\mathrm{T}} = 4\%$. The drivetrain dynamics (2) in closed-loop with the controller (5) are incorporated to represent the wind turbine. Ten wind turbines within $P_{\mathrm{g,rated}} \in [5, 25]$ MW range are considered and the inverse of (1b) is made use of to obtain $R = D/2$ from the specified $P_{\mathrm{g,rated}}$, e.g., to compute TSR and the optimal mode gain $K$.

Their estimator gains are subsequently obtained using (22), in which $\omega_0 = 25$ rad/s and $\zeta_0 = 1$ are chosen and, as will be shown later, result in satisfactory estimator performance. The $J$ values derived from (1c) for the selected $P_{\mathrm{g,rated}}$ range and are subsequently substituted to (22b) to adjust $L_2$[5]. For the constant gain case, the $L_2$ computed for $P_{\mathrm{g,rated}} = 5$ MW is considered for all turbines. No noise is assumed for $\omega_{\mathrm{r}}$ measurements for simplicity's sake in this demonstration; nevertheless, similar conclusions can be derived under noisy measurements.

Figure 5 summarizes the key statistical results of the simulations, being the absolute means ($|\mu_{(\bullet)}|$) and standard deviations ($\sigma_{(\bullet)}$) of the rotor speed, aerodynamic torque, and aerodynamic power estimation errors, $\omega_{\mathrm{r}} - \hat{\omega}_{\mathrm{r}}$ and $T_{\mathrm{r}} - \hat{T}_{\mathrm{r}}$, and $P_{\mathrm{r}} - \hat{P}_{\mathrm{r}}$ respectively. In general, it is observed from the figure that, compared with the tailored gain case, the use of constant gain deteriorates the absolute means and standard deviations as the power rating increases. However, an exception applies for $\sigma_{T_{\mathrm{r}} - \hat{T}_{\mathrm{r}}}$ and $\sigma_{P_{\mathrm{r}} - \hat{P}_{\mathrm{r}}}$ where similar results are depicted for both cases.

---

[4]That is, $s^2 + 2\zeta_0\omega_0 s + \omega_0^2 = 0$. Applying Forward Euler discretization to this equation gives (19).

[5]Alternatively, one may also use the previously obtained $D$ followed by a substitution to (1a).

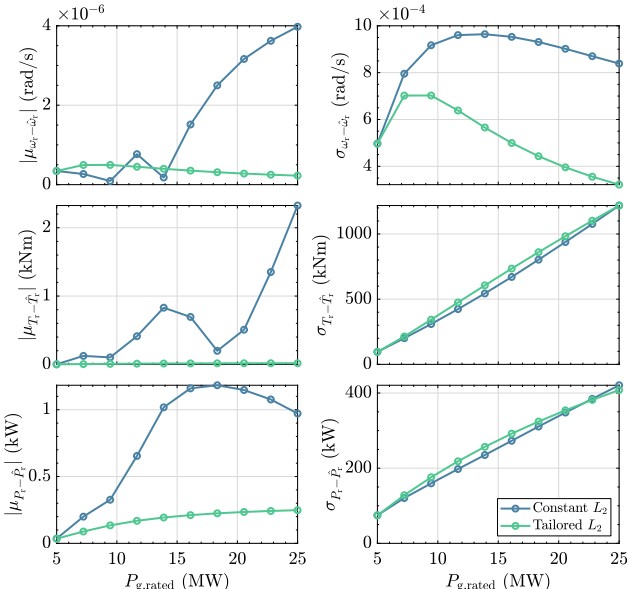

**Figure 5.** Statistical assessment results of the constant (blue) and tailored (green) Luenberger estimator's $L_2$ gain based on the turbine's power rating (and inertia). Absolute errors of the rotor speed, aerodynamic torque, and aerodynamic power estimates (left column) of the constant $L_2$ strategy tend to be much higher than the tailored gain. Significant difference in the standard deviation of these errors (right column) is only shown for the the rotor speed estimation, whereas those of the aerodynamic torque and power are comparable. These results imply that appropriate gain adjustment based on the turbine's power rating is imperative.

The observation from the above demonstration motivates the need to set a new standard, i.e., by employing $L_2$ tailoring based on the power rating and consequent drivetrain inertia of the considered turbines, which provides a convenient means to calibrate state estimation-based aerodynamic power estimation. As will be shown later, such a gain-tailoring—and more importantly, the state estimation-based aerodynamic power estimator—leads to faster (less phase lag) and more noise-resilient wind speed estimation.

### 4.3 Aerodynamic Power Estimation Techniques Comparison

With the numerical derivative and state estimation approaches to estimate aerodynamic power already presented in the previous sections at hand, this section is now dedicated to comparing both methods. To this end, simulations with the same turbulent wind setting in Section 4.2.2 are run, where a wind turbine of $P_{\text{g,rated}} = 15$ MW, representing a 'mid-sized' turbine in the considered turbine range, is utilized. In addition, noisy rotor speed measurements are assumed, with $\sigma_{v_{\omega_{\text{r}}}}^2 = 10^{-6}$ rad$^2$/s$^2$.

Two strategies in obtaining $\hat{P}_{\text{r}}$ are compared:

1. Using filtered derivative $F_{\text{nd,d}}$ introduced in Section 4.1 to obtain $\hat{\dot{\omega}}_{\text{r}}$, followed by its substitution to (8), including $\omega_{\text{r}}$ and $P_{\text{g}}$ measurements with known $J$ according to Assumption 5. A time constant of $\tau = 0.5$ s is selected as it is considered a good trade-off between noise correlation, quality of the derivative, and noise amplification limitation;




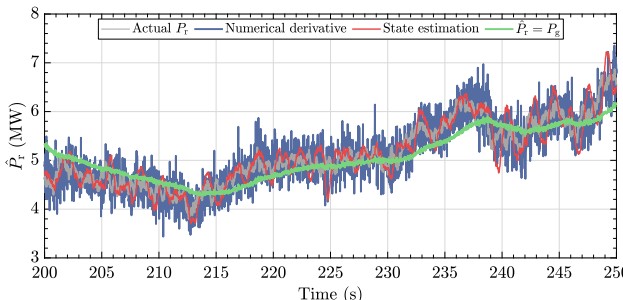

**Figure 6.** Comparison of aerodynamic power estimation methods. It is shown that the estimate obtained via the state estimation-based technique (red lines) has much less noise compared with that of the numerical derivative one (blue lines) while still maintaining high estimation accuracy. Also depicted is the aerodynamic power estimate determined only using the generated power (green lines), which shows the worst-case phase lag with respect to the other methods, demonstrating the loss of information if rotor acceleration information is absent.

2. Directly retrieving $\hat{P}_r$ by state estimation method explained in Section 4.2 by multiplying $\hat{\omega}_r$ and $\hat{T}_r$ (see Remark 3). The gain $\boldsymbol{L}$ is computed by setting $\omega_0 = 25 \text{ rad/s}$ and $\zeta_0 = 1$ as used in the previous section.

Figure 6 depicts the time series results of the simulation, where, for clarity, only chosen for the timestamp $t = 200 - 250$ s. In the figure, the actual aerodynamic power as a ground truth is indicated by the gray line. It can be seen that the aerodynamic power estimation result for the state estimation-based method excels that of the numerical derivative one in terms of less noise propagation and phase lag.

Note that, for the latter, increasing $\tau$ will result in less noise but increasing phase lag (Østergaard et al., 2007) as this will diminish and deteriorate the $J\omega_r\dot{\omega}_r$ estimate such that $\hat{P}_r \approx P_g$ (i.e. information will be lost). The case where $J\omega_r\dot{\omega}_r = 0$ such that $\hat{P}_r = P_g$ is demonstrated by the green line, which evidently shows a phase lag with respect to those where the information of $J\omega_r\dot{\omega}_r$ is made available. It is concluded, therefore, that for the power balance REWS estimation scheme, the state estimation-based aerodynamic power estimator is to be used for the remainder of this paper. Note that the performance of the state estimation method can be improved by further tuning of $\omega_0$ and $\zeta_0$.

## 5 Wind Speed Estimate Solver

Having a good estimate of the aerodynamic power is crucial for the second component of the overall power balance wind speed estimation scheme, which solves the effective wind speed estimate (see Fig. 2). Alluded to earlier in Section 3.3 and shown in Fig. 3, the two manners in which such a solver can be designed are detailed in the following sections. Section 5.1 discusses the continuous solver, where the linear state-space derivation of the solver is done, followed by frequency-domain analysis. Then the stability of the solver in the discrete-time domain is discussed. Later, in Section 5.2, the iterative single-step algorithm is proposed as a promising alternative to the former.



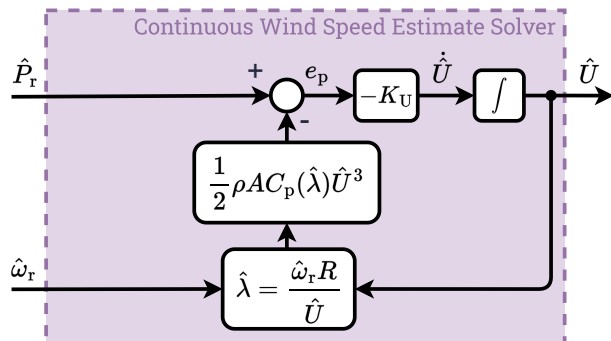

**Figure 7.** Internal structure of the continuous wind speed estimate solver.

## 5.1 Continuous Solver

This section presents an analysis of the continuous manner to solve the wind speed estimate, given that the aerodynamic power estimate and rotor speed measurements are provided. Figure 7 depicts the internal structure of this continuous wind speed estimate solver and is elaborated in the following.

As laid out in Section 3.2, asymptotically minimizing the estimation error term $e_\mathrm{p}$ returns the wind speed estimate $\hat{U}$, which converges to the actual wind speed over time per the definition in (6). Such an integration operation enables the wind speed estimate solver to be written as the following state transition equation

$$\dot{\hat{U}}(t) = -K_\mathrm{U} e_\mathrm{p}(t), \tag{23}$$

with the integrator gain

$$K_\mathrm{U} = \frac{\kappa}{P_\mathrm{g,rated}}, \tag{24}$$

determining the convergence rate. The notation $\kappa \in \mathbb{R}^+$ is a constant and the rated generator power $P_\mathrm{g,rated} \in \mathbb{R}^+$ is used to convert $e_\mathrm{p}$ from wattage into the per-unit (p.u.) system.

As $e_\mathrm{p}$ has a nonlinear analytic definition (7), the wind speed estimate solver in the continuous time is represented by the following nonlinear dynamics

$$\mathcal{S} : \begin{cases} \dot{x}(t) &= f(x(t), \boldsymbol{u}(t)), \\ y(t) &= g(x(t)), \end{cases}$$

with

$$x = \hat{U}, \quad \boldsymbol{u} = \left[\hat{P}_\mathrm{r}, \hat{\omega}_\mathrm{r}\right]^\top, \text{ and } \quad y = \hat{U},$$

as its state, input, and output vectors, respectively.



To proceed with the linear analysis, the first-order Taylor expansion of $\mathcal{S}$ is derived, resulting in the following linear state-space system

$$
\begin{aligned}
\dot{x}(t) &= A x(t) + \boldsymbol{B}\boldsymbol{u}(t), \\
y(t) &= C x(t),
\end{aligned}
\tag{25}
$$

with the state, input, and output matrices defined by the following Jacobians

$$
\begin{aligned}
A &= \left.\frac{\partial f}{\partial x}\right|_{(\bar{x},\bar{\boldsymbol{u}},\bar{y})} = -\frac{K_\mathrm{U}}{2}\rho A_\mathrm{r}\bar{U}^2\left(3C_\mathrm{P}\left(\bar{\lambda}\right) - \bar{\lambda}\frac{\partial C_\mathrm{P}}{\partial\hat{\lambda}}\right), \\
\boldsymbol{B} &= \left.\frac{\partial f}{\partial\boldsymbol{u}}\right|_{(\bar{x},\bar{\boldsymbol{u}},\bar{y})} = -K_\mathrm{U}\left[1 \quad -\frac{1}{2}\rho A_\mathrm{r}R\bar{U}^2\frac{\partial C_\mathrm{P}}{\partial\hat{\lambda}}\right], \\
C &= \left.\frac{\partial g}{\partial x}\right|_{(\bar{x},\bar{\boldsymbol{u}},\bar{y})} = 1,
\end{aligned}
\tag{26}
$$

respectively.

Given the linearized dynamics above, it becomes compelling to examine the stability properties of the linear system. To this end, the next subsections provide frequency-domain stability and discretization-method analysis utilizing the above-derived linear system.

### 5.1.1 Frequency-Domain Stability Analysis

In the previous section, the nonlinear dynamics of the continuous solver have been described, followed by their linear state-space rendition. Here, the stability of the solver is assessed via pole location investigation. Solving for $\boldsymbol{G}(s) = Y(s)/\boldsymbol{U}(s) = C(s-A)^{-1}\boldsymbol{B}$, one obtains the multiple-input single-output transfer matrix formulation of the state-space (25) as follows

$$
\boldsymbol{G}(s) = \frac{\boldsymbol{N}}{D(s)}.
\tag{27}
$$

The notations $\boldsymbol{N} = \boldsymbol{B}$ and $D(s)$ are the respective numerators and denominator of the transfer functions above, with the former being a constant gain vector, hence the independence from $s$. The latter is of interest, especially with regard to stability analysis.

The denominator of the transfer functions is

$$
D(s) = s + \underbrace{\frac{K_\mathrm{U}}{2}\rho A_\mathrm{r}\bar{U}^2\left(3C_\mathrm{P}\left(\bar{\lambda}\right) - \bar{\lambda}\frac{\partial C_\mathrm{P}}{\partial\hat{\lambda}}\right)}_{p},
\tag{28}
$$

which, in order to guarantee stability, left half-plane pole location $p < 0$ must be satisfied such that

$$
p = -\underbrace{\frac{K_\mathrm{U}}{2}\rho A_\mathrm{r}\bar{U}^2}_{p_c}\left(\underbrace{3C_\mathrm{P}\left(\bar{\lambda}\right)}_{p_a} - \underbrace{\bar{\lambda}\frac{\partial C_\mathrm{P}}{\partial\hat{\lambda}}}_{p_b}\right) < 0,
\tag{29}
$$

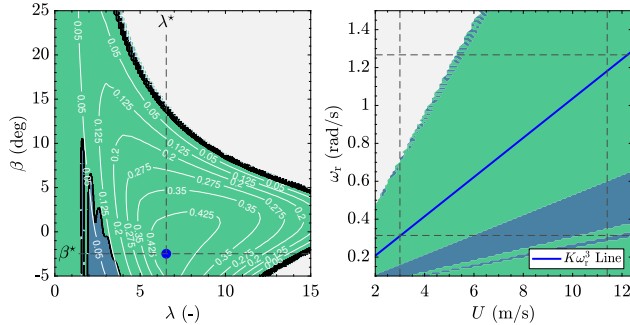

**Figure 8.** Stability region of the continuous wind speed estimate solver in the continuous time $\boldsymbol{G}(s)$, where the condition $p < 0$ in (29) is satisfied (indicated in green, otherwise in blue). The left subfigure shows the different $C_\mathrm{p}$ levels for the NREL-5MW turbine (white solid lines), with $C_\mathrm{p}^\star$ indicated by the blue dot. The right subfigure shows the mapping of the stability region in terms of wind speed $U$ and rotor speed $\omega_\mathrm{r}$ for the fine pitch angle $\beta^\star$. The solid blue line is the partial-load regime, where the $K\omega_\mathrm{r}^3$ controller is active. Also shown for completeness are the black dashed lines, indicating the lower and upper bounds for the rotor speed, as well as cut-in and rated wind speeds.

with $p_c > 0$. The inequality (29) requires $p_a - p_b > 0$ to hold to ensure the pole stays within the left-half plane. This condition,
which can be rewritten as

$$p_a > p_b \quad \Leftrightarrow \quad \frac{3}{\bar{\hat{\lambda}}} C_\mathrm{p}\left(\bar{\hat{\lambda}}\right) > \frac{\partial C_\mathrm{p}}{\partial \hat{\lambda}}, \tag{30}$$

is a well-known condition for global asymptotic stability in the (Improved) Immersion and Invariance wind speed estimator works, e.g., Ortega et al. (2013) and Liu et al. (2022).

With the stability expression of (29)/(30) at hand, a stability map at different operating conditions is made. For this purpose, the $C_\mathrm{p}$ table of the NREL-5MW is taken. Figure 8 depicts the resulting stability region of the continuous wind speed estimate solver. The stable region is shown in green, whereas the unstable region is in blue. The left subfigure illustrates how the $C_\mathrm{p}$ contour is divided based on whether (29)/(30) is satisfied. The blue dot shows the location of $C_\mathrm{p}^\star$, corresponding to the fine pitch $\beta^\star$ (horizontal dashed line) and design TSR $\lambda^\star$ (vertical dashed line). The operating conditions at the $\beta^\star$ line are mapped on the right subfigure, resulting in the stability region representation in terms of $U$ and $\omega_\mathrm{r}$. The blue solid line represents $\lambda^\star$ or $K\omega_\mathrm{r}^3$ line, where maximum power extraction occurs. As can be seen, the continuous wind speed estimate solver is stable for a large part of the turbine operational domain and, more importantly, along the optimal $K\omega_\mathrm{r}^3$ line where the turbine operates in partial-load in the steady state.

However, as far as practicality is concerned, the effects of discrete-time implementations should extend the above continuous-time analysis. A discretization method, for instance, might not preserve the stability properties obtained in the continuous-time domain (Åström and Wittenmark, 2011). Moreover, different estimator gain $\kappa$ values influence the stability region. These aspects are the main concerns in the next section.





### 5.1.2 Solver Discretization and Instability

In this section, the stability of the continuous wind speed estimate solver (27) in the discrete time is discussed. The three most common discretization methods are considered as follows (Åström and Wittenmark, 2011)

1. The Forward Euler (FE) method, the $s$-to-$z$-domain transformation of which is as follows

$$s'_{\text{FE}} = \frac{z-1}{h}. \tag{31}$$

2. The Backward Difference (BD) method, with the following transformation

$$s'_{\text{BD}} = \frac{z-1}{zh}. \tag{32}$$

3. The Tustin (TU) method, with

$$s'_{\text{TU}} = \frac{2}{h}\frac{z-1}{z+1}. \tag{33}$$

as the corresponding $s$-to-$z$-domain transformation.

Substituting any of the above discretization methods into $\boldsymbol{G}(s)$ results in the following general discrete-time approximation representation

$$\boldsymbol{G}(s = s') = \boldsymbol{H}(z) = \frac{\boldsymbol{P}(z)}{Q(z)}, \tag{34}$$

with $\boldsymbol{P}(z)$ denoting the numerators and $Q(z)$ the denominator of the discrete-time transfer function $\boldsymbol{H}(z)$.

Similar to its continuous-time counterpart, discrete-time stability analysis only focuses on the poles of the system, i.e. whether they are within the unit disc, such that in the following study, only $Q(z)$ is of interest (Åström and Wittenmark, 2011). Explicit representations for $\boldsymbol{H}(z)$, obtained by the discretization methods (31)-(33), are, therefore, provided below and their corresponding stability condition derivation follows

1. Using $s = s'_{\text{FE}}$, $\boldsymbol{G}(s)$ becomes

$$\boldsymbol{H}_{\text{FE}}(z) = \frac{\boldsymbol{N}h}{z + K_{\text{U,h}}\frac{1}{2}\rho A_{\text{r}}\bar{\hat{U}}^2(p_a - p_b) - 1}, \tag{35}$$

with $K_{\text{U,h}} = hK_{\text{U}}$ (or, similarly, $K_{\text{U,h}} = h\kappa/P_{\text{g,rated}}$). From (35), the following inequality must hold for stability to hold

$$\left| 1 - K_{\text{U,h}}\frac{1}{2}\rho A_{\text{r}}\bar{\hat{U}}^2(p_a - p_b) \right| < 1. \tag{36}$$

2. For the BD method, $s = s'_{\text{BD}}$ is used and the following discrete transfer function is obtained

$$\boldsymbol{H}_{\text{BD}}(z) = \frac{(\boldsymbol{N}h)z}{\left(1 + K_{\text{U,h}}\frac{1}{2}\rho A_{\text{r}}\bar{\hat{U}}^2(p_a - p_b)\right)z - 1}, \tag{37}$$





with the following condition for stability

$$
\left| \frac{1}{1 + K_{\mathrm{U,h}} \frac{1}{2} \rho A_{\mathrm{r}} \bar{\hat{U}}^2 (p_a - p_b)} \right| < 1 .
\tag{38}
$$

3. Under the TU discretization, $s = s'_{\mathrm{TU}}$ casts $\boldsymbol{G}(s)$ into


$$
\boldsymbol{H}_{\mathrm{TU}}(z) = \frac{\frac{\boldsymbol{N} h}{2}(z + 1)}{z + K_{\mathrm{U,h}} \frac{1}{4} \rho A_{\mathrm{r}} \bar{\hat{U}}^2 (p_a - p_b)(z + 1) - 1} ,
\tag{39}
$$

the stability of which is determined by the following inequality

$$
\left| \frac{1 - K_{\mathrm{U,h}} \frac{1}{4} \rho A_{\mathrm{r}} \bar{\hat{U}}^2 (p_a - p_b)}{1 + K_{\mathrm{U,h}} \frac{1}{4} \rho A_{\mathrm{r}} \bar{\hat{U}}^2 (p_a - p_b)} \right| < 1 .
\tag{40}
$$

Note that the stability properties may now be influenced by the choice of sampling time $h$ and gain $\kappa$ (as $P_{\mathrm{g,rated}}$ is constant). Nevertheless, to illustrate better whether alterations in the stability region occur after discretization takes place, the stability

conditions (36), (38), and (40) are plotted in a similar manner as Fig. 8, as explained in the following text. Also considered are two arbitrary operating points (OPs) along the $K\omega_{\mathrm{r}}^3$ line for a low and high partial-load wind speed being

$$
\text{OP 1:} \begin{cases} \bar{x} = \bar{y} = \bar{\hat{U}} = 4.5\ \mathrm{m/s} , \\ \bar{\boldsymbol{u}} = \left[ \bar{\hat{P}}_{\mathrm{r}}, \bar{\omega}_{\mathrm{r}} \right]^\top = [3.263 \times 10^5\ \mathrm{W}, 0.469\ \mathrm{rad/s}]^\top , \end{cases}
$$

and

$$
\text{OP 2:} \begin{cases} \bar{x} = \bar{y} = \bar{\hat{U}} = 10.5\,\mathrm{m/s} , \\ \bar{\boldsymbol{u}} = \left[ \bar{\hat{P}}_{\mathrm{r}}, \bar{\omega}_{\mathrm{r}} \right]^\top = [4.145 \times 10^6\ \mathrm{W}, 1.094\ \mathrm{rad/s}]^\top , \end{cases}
$$

respectively, computed using the NREL-5MW properties.

First, the stability region for the FE method, with $h = 1/50\,\mathrm{s}$—constant for all evaluations, is examined. Figure 9 depicts the resulting stability assessment, which illustrates the deterioration of the FE method's stability region as $\kappa$ increases, affecting the partial-load operations (blue solid line), e.g. OP 2 (red dot). Although not shown in the figure, even higher $\kappa$ may affect low-wind speed operations. This observation, therefore, concludes that one's choice of discretization method results in a perfor-

mance limitation of the wind speed estimate solver in terms of an existing 'upper bound' for $\kappa$'s magnitude. A compromise can be made, nevertheless, to improve the FE method's stability by increasing sampling frequency (i.e., lowering $h$) proportional to the increase in $\kappa$ to maintain constant $K_{\mathrm{U,h}}$. That said, increasing the sampling frequency does not eliminate the presence of a $\kappa$ 'upper bound,' not to mention the extent to which such a frequency can be increased is practically limited as a result, e.g., hardware capabilities. Therefore, a more feasible solution is to adopt different discretization methods while leaving the

sampling frequency unchanged.

Figure 10 makes clear that the BD method, in contrast to the FE method, does not result in the change of stability characteristics of the continuous system into the discrete system for the considered $\kappa$ values. Remarkably, if $\kappa$ is further increased,

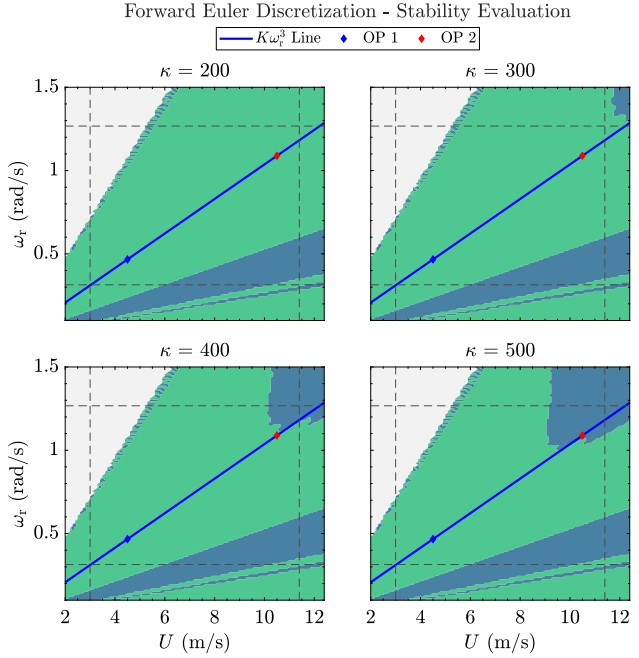

**Figure 9.** Unstable region (blue area) growth of $\boldsymbol{H}_{\mathrm{FE}}(z)$ with increasing $\kappa = \{200, 300, 400, 500\}$ using $h = 1/50$ s of sampling time and the Forward Euler discretization method.

the pole of the discrete-time system $\boldsymbol{H}_{\mathrm{BD}}(z)$ moves away even more from the edge of the unit circle (as implied in (38), theoretically leading to increased stability for increasing gains).

Similar to the BD method, the TU method gain increase will lead the discrete-time pole to move closer to the origin, but under the same gain, that of the TU method will be closer to the origin than the BD method. This is because with the TU method, higher gains simultaneously lower the numerator and increase the denominator of the discrete pole, as can be implied from (40). Regardless, for the considered $\kappa$, the stability region of the TU-discretized system $\boldsymbol{H}_{\mathrm{TU}}(z)$ is identical to that of $\boldsymbol{G}(s)$. It is worth noting, therefore, that although Fig. 10 shows the stability characteristics of $\boldsymbol{H}_{\mathrm{BD}}(z)$ for the given gains, that of $\boldsymbol{H}_{\mathrm{TU}}(z)$ would give identical representations; therefore, no dedicated figure is provided for the latter for brevity.

To summarize, the BD and TU methods are the preferred discretization techniques for discrete-time implementations of the continuous wind speed estimate solver in that the stability condition from the continuous-time system is preserved. Given present-day computational resources and readily available discretization methods in popular software packages, the selection for each of the methods is inconsequential from an implementation perspective. Therefore, later on in Section 5.3, a time-series numerical comparison is performed to determine the most suitable approach, also including the iterative single-step solver explained in the next section.

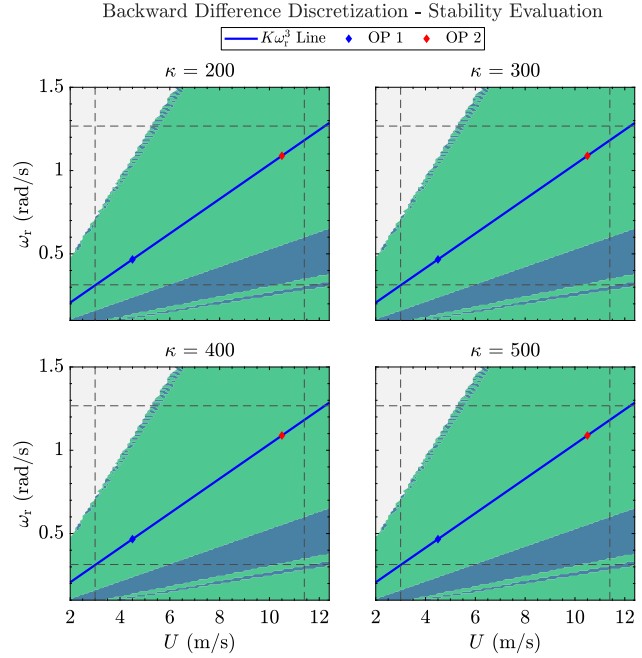

**Figure 10.** Stability region of $\boldsymbol{H}_{\mathrm{BD}}(z)$ with increasing $\kappa = \{200, 300, 400, 500\}$ using $h = 1/50$ s of sampling time and the Backward Difference (BD) discretization method. Identical graphical evaluation is obtained for the Tustin-discretized system $\boldsymbol{H}_{\mathrm{TU}}(z)$ for the selected gains.

## 5.2 Iterative Single-Step Solver

Besides the continuous wind speed estimate solver in the previous section, iterative numerical methods can also be employed to solve for the wind speed estimate. A well-known iterative method for this purpose is that of the Newton-Raphson, such as used in the work of Van Engelen and Van der Hooft (2003) and Boukhezzar and Siguerdidjane (2011). This iterative algorithm finds the roots of a function given an initial guess and makes use of the function's gradient. The reliance on such a gradient, however, adds an additional layer of complexity in this case in that an extra look-up table other than that for the $C_{\mathrm{p}}$ table is needed. Moreover, the use of this extra look-up table would increase the computational burden per iteration within the algorithm. Luckily, the definition of the wind speed estimate, being an integration of the estimation error $e_{\mathrm{p}}$ over time multiplied by a gain $K_{\mathrm{U}}$ can be straightforwardly adopted in an iterative manner in such a way that $\hat{U}$ can be obtained in a single time step. Algorithm 1 describes the proposed iterative single-step wind speed estimate solving method.

First, the iterative single-step algorithm computes the TSR estimate based on $\hat{\omega}_{\mathrm{r}}(k)$, available through the aerodynamic power estimator in Section 4, and wind speed estimate $U_i$ at the $i$-th iteration, namely $\lambda_i$. Having $\lambda_i$, the $U_i$-dependent aerodynamic power estimate can then be computed, which is used to obtain the aerodynamic power error along with $\hat{P}_{\mathrm{r}}(k)$, available from the aerodynamic power estimator. Normalized by $P_{\mathrm{g,rated}}$, this error (denoted $e_{\mathrm{p,norm},i}$), is then used to update the $i+1$-th wind speed estimate $U_{i+1}$. These steps are then repeated until the relative wind speed estimate error or the absolute, normalized,



---

**Algorithm 1** Iterative single-step method for solving wind speed estimate

---

**Require:** $\hat{P}_r(k), \hat{\omega}_r(k), \hat{U}(k-1), i_{\max}, \epsilon_U, \epsilon_p$

**Ensure:** $U \approx \hat{U}(k)$

1: $i \leftarrow 1$

2: $U_i \leftarrow \hat{U}(k-1)$

3: **repeat**

4:      $\lambda_i \leftarrow \dfrac{\hat{\omega}_r(k)R}{U_i}$

5:      $e_{p,\mathrm{norm},i} \leftarrow \dfrac{\hat{P}_r(k) - \frac{1}{2}\rho A C_p(\lambda_i)U_i^3}{P_{g,\mathrm{rated}}}$

6:      $U_{i+1} \leftarrow U_i + e_{p,\mathrm{norm},i}$

7:      $U_{\mathrm{old}} \leftarrow U_i$

8:      $i \leftarrow i+1$

9: **until** $(i \geq i_{\max})$ or $\left( \left| \frac{(U_i - U_{\mathrm{old}})}{U_i} \right| \leq \epsilon_U \right)$ or $(|e_{p,\mathrm{norm},i}| \leq \epsilon_p)$

10: $\hat{U}(k) \leftarrow U_i$

11: **return** $\hat{U}(k)$

---

aerodynamic power error falls within the tolerance bound $\epsilon_U \in \mathbb{R}^+$ or $\epsilon_p \in \mathbb{R}^+$, respectively. Otherwise, the algorithm stops until the maximum allowed iteration $i_{\max} \in \mathbb{Z}^+$ is reached. Finally, the algorithm outputs the wind speed estimate $\hat{U}(k) = U_i$ from the last iteration. Note that, compared to the continuous wind speed estimate solver, the iterative method here employs

$\kappa = 1$. Later, it will be shown that setting such a unity gain is sufficient to achieve fast convergence.

### 5.3 Wind Speed Estimate Solvers Comparisons

With the continuous and iterative single-step solvers explained in the previous sections, this section now compares both methods numerically. The optimal wind speed estimate solver is then picked and combined with the state estimation-based aerodynamic power estimator, as discussed in Section 4.2. To this end, the same simulation setup in Section 4.3 is considered.

First, the performance of the different continuous wind speed estimate solvers, discretized under the FE, BD, and TU methods, are compared. The estimator gain is chosen to be $\kappa = 400$, which is a stable gain, especially for the FE discretization at the considered operating condition ($U_h = 7.5\,\mathrm{m/s}$, under $I_T = 4\%$). Figure 11 shows the simulation results, which is focused on $t = 210 - 230$ s timestamp for clarity. As shown in the figure, the BD- and TU-discretized wind speed estimate solvers show identical estimation performance, which is not the case in their FE-discretized counterpart. The unstable-like oscillations

occurring in the beginning, middle, and end of the sequence of interest of the FE method are likely resulting from nonlinearity effects in combination with frequency folding or aliasing (Åström and Wittenmark, 2011).

Figure 12 depicts the time-series comparison between the BD-discretized continuous solver and the iterative method. With the same tuning parameters as the previous simulation for the former, the latter is configured with $i_{\max} = 5$ and $\epsilon_U = \epsilon_p = 0.01$, which are as a good trade-off between accuracy and speed of the estimation. As evidently shown, both solvers demonstrate



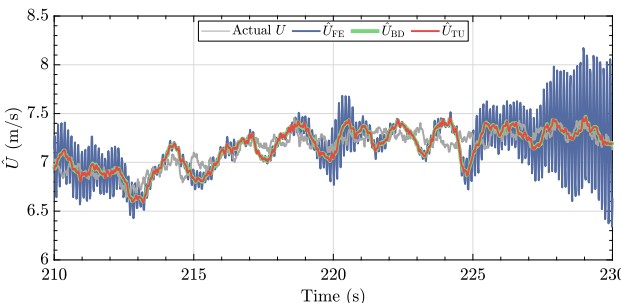

**Figure 11.** Actual wind speed $U$ and wind speed estimates of the continuous wind speed estimate solver under different discretization methods $\hat{U}_{(\bullet)}$. The FE-discretized solver occasionally shows high-frequent oscillatory behavior, potentially due to combined nonlinearity and aliasing effects, which is not the case for that of the BD and TU methods for the chosen estimator gain $\kappa = 400$.

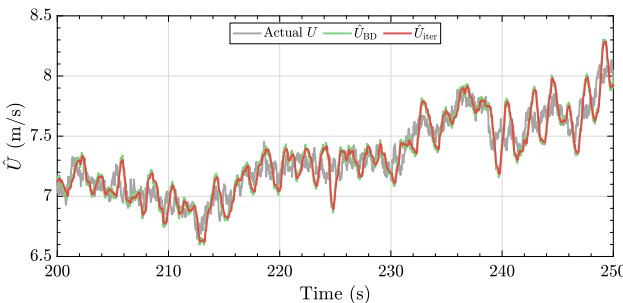

**Figure 12.** Actual wind speed $U$ and wind speed estimates of the continuous and iterative single-step wind speed estimate solvers, the former of which is discretized with the BD method. Identical responses indicate that the accuracy of the wind speed estimate solvers depends, to a large extent, on the aerodynamic power (and rotor speed) estimation quality.

similar performance and accuracy for the presented case. However, in different scenarios, such as more noisy aerodynamic power and rotor speed estimates, both solvers may show disparities.

Noisy inputs would require the continuous solver's integrator gain to be lowered, such that high-frequency components in the input signals are attenuated but potentially resulting in lagged estimates. For the iterative solver, noisy inputs would increase the computational cost in terms of the higher number of iterations to converge to a solution.

Fortunately, as the role of the aerodynamic power estimation (with a bonus of rotor speed filtering) and wind speed estimate solving are decoupled in this study, the task to ensure low-noise and accurate wind speed estimate relies mostly upon the former's subcomponent tuning. This way, few iterations and strict error tolerances of the iterative algorithm can be maintained. Such a condition also benefits the continuous solver's wind speed estimate; however, as it requires a high gain to maintain good estimation quality, it is still prone to having the above-mentioned sampled-time system artifacts.



Hence, for the final and optimal design of the power balance wind speed estimation in this work, the combination of the state estimation-based aerodynamic power estimator and iterative single-step wind speed solver is chosen and is evaluated next in higher-fidelity simulations.

## 6   Higher-Fidelity Simulation Setup and Results

With the power balance wind speed estimation design finalized, this section covers the higher-fidelity validation of the proposed
algorithm. The details of the simulation setup are covered in Section 6.1, and the validation results are discussed in Section 6.2.

### 6.1   Simulation Setup

The simulation setup for the higher-fidelity wind speed estimator validation in this work uses the open-source simulation code OpenFAST v3.5.3 (Jonkman et al., 2024), the development of which is led by the National Renewable Energy Laboratory (NREL). OpenFAST couples several nonlinear aero-hydro-servo-elastic computational modules by which realistic and complex
wind turbine dynamic responses can be simulated with high accuracy. For the validation purposes of this work, the AeroDyn, ServoDyn, ElastoDyn, and InflowWind modules of OpenFAST are used. Concerning the degrees of freedom (DOFs) of the simulated wind turbines, the following are activated:

- generator;

- drivetrain rotational-flexibility;

- first and second flapwise blade modes;

- first edgewise blade mode;

- first and second fore-aft tower modes; and

- first and second side-side tower modes;

Note that the drivetrain rotational-flexibility DOF is turned off when a direct-drive wind turbine is simulated.
Two wind turbines representing the respective low- and high-power ratings are simulated, namely the NREL-5MW and the IEA-22MW turbines, as introduced in Section 2.2. In contrast to using equal $C_\mathrm{p}$ tables in the previous analysis sections, using the reference wind turbine models in OpenFAST leads to simulating the aerodynamic properties of the respective turbines.

    With regard to the wind profile, Kaimal turbulent wind cases are considered for both turbines, with $U_\mathrm{h} = 7.5$ m/s and $I_\mathrm{T} = \{4, 12\}\%$, generated using TurbSim (Jonkman, 2014), and used as input for the aforementioned InflowWind module. The
simulations are run for 1060 s, in which the first 60 s is excluded to remove computational transients from the evaluation.

    The power balance wind speed estimator employs the state-estimation-based method for the aerodynamic power estimation (Section 4) with its gain $L$ computed using $\omega_0 = 5.75$ rad/s and $\zeta_0 = 4.5$. Note that, compared to the initial low-fidelity simulations in Section 4.2.2, the lower frequencies and higher dampings of the estimator are chosen and considered to be a



good compromise between noise filtering and good performance in the higher-fidelity settings. Better performance might be
attained by the incorporation of more systematic tuning methods that are able to find the optimal gain via cost minimization
(e.g., mean and variance of the wind speed estimate error), such as Bayesian optimization (Mulders et al., 2020) or genetic
algorithms (Lara et al., 2024). With regard to the measurement noise, that of the rotor speed $v_{\omega_\mathrm{r}}$ is assumed to have variance
of $\sigma^2_{v_{\omega_\mathrm{r}}} = 10^{-5} \ \mathrm{rad}^2/\mathrm{s}^2$, i.e. one order of magnitude higher than that used in the low-fidelity simulations in Section 4.3.

Constant and tailored gain settings are considered, the first one being computed using the NREL-5MW's LSS-equivalent
inertia for both turbines, similar to what was done in Section 4.2.2, whereas the second one is tailored on the actual LSS-
equivalent inertia of the simulated turbines (see Table 1). For the two turbines considered in this section, the chosen $\omega_0$ and $\zeta_0$
result in stable aerodynamic power estimators for both constant and tailored gain cases.

For the iterative single-step wind speed estimate solver, $i_\mathrm{max} = 10$ and $\epsilon_\mathrm{U} = \epsilon_\mathrm{p} = 0.005$ are considered. Despite the tighter
convergence bounds compared to the low-fidelity simulations in Section 5.2, the wind speed estimate in the simulations of this
section only requires one iteration in average to solve.

The proposed method is also compared with an existing wind speed estimator, namely the Immersion and Invariance (I&I)
REWS estimator, based on the work of Liu et al. (2022), the brief derivation of which is provided in Appendix A. This estimator
is based on torque balance drivetrain dynamics, in which the internal model estimates the rotor speed and aims to minimize its
difference from the actual measurements. Resultingly, the error compensation by a proportional-integral (PI) structure gives the
wind speed estimate of this scheme. For both turbines in the low turbulence scenario, the I&I estimator gains are $K_\mathrm{p} = 15$ m
and $K_\mathrm{i} = 3.5$ m/s, whereas for the high turbulence case, the gains are retuned to be $K_\mathrm{p} = 25$ m and $K_\mathrm{i} = 5$ m/s to match
the performance of the proposed method. Note that there have not been any studies yet in systematic tuning across a range
of turbine sizes for the I&I approach, to the best of the authors' knowledge. Thus, the above PI gains are tuned heuristically
and equally for NREL-5MW and IEA-22MW. The next section covers the results from the OpenFAST simulations for the
aforementioned settings.

## 6.2 Results

The time series evaluation of the power balance wind speed estimator's performance is provided in this section for the con-
sidered reference turbines, NREL-5MW and IEA-22MW, where for each turbine. Later on, the statistical assessments of the
numerical simulation results are provided at the end of this section.

Figure 13 depicts the time series results of the wind speed estimation for the NREL-5MW. For both the turbulent cases, it
is shown that both methods are able to capture the slow-varying component of the actual wind speed well, with the proposed
method showing less noisy results compared to I&I. This is mainly because the latter method, by default, does not contain
any noise filtering feature, which can be added by low-pass filtering, for instance. Additional filtering would lead to additional
phase lags and, thus, a slower wind speed estimation. The proposed estimator, thus, demonstrates superior noise handling
capabilities over I&I, which obviates the need for additional filtering. However, as such a modification is not the main focus of
this work, the original I&I structure is retained.



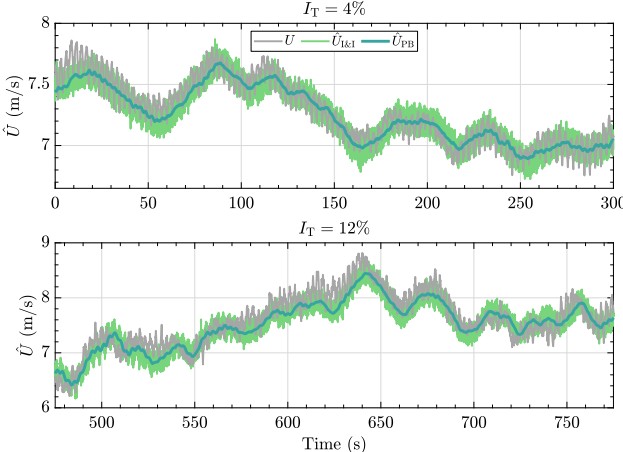

**Figure 13.** Wind speed estimation time series for the NREL-5MW wind turbine. The proposed method (blue line) gives a smooth, less noisy estimate compared to the I&I (green line) for both turbulent cases. The low-frequent component of the wind speed is captured where biased estimation occasionally occurs, potentially due to an inaccurate $C_p$ table utilized in both estimators and an absence of dynamic inflow modeling in the estimators. The actual REWS, as outputted by OpenFAST, is shown in the gray line.

An interesting behavior worth paying attention to from both wind speed estimation methods is the somewhat equal biases with respect to the actual wind speed. This is evident at the beginning of the low turbulent wind case and at 525 s and 625 s of the high turbulent case. Such biases are potentially coming from the inaccuracy in the $C_p$ table (Brandetti et al., 2022), which was generated by steady-state simulations, which might not necessarily be accurate during transients. Additionally, the absence of dynamic inflow effects in the estimator model could play a role in the appearance of such estimation biases (Knudsen and Bak, 2013).

Having the estimation results for the NREL-5MW presented, that of the IEA-22MW is showcased. The main goal of the simulations with a larger turbine is to validate the performance differences when constant and tailored estimator gains are used, the proof-of-concept of which was shown in the low-fidelity simulations of Section 4.2.2. Figure 14 depicts the time series results of the wind speed estimation for the IEA-22MW wind turbine. For the constant gain case, the $L_2$ aerodynamic power estimator gain is based on the one tuned for the NREL-5MW turbine, whereas that of the tailored gain case is determined based on the inertia of the IEA-22MW turbine using (22b). Evident in the figure is the better performance of the proposed method under tailored $L_2$ compared to the constant $L_2$—note the displayed lagged behavior of the latter. With respect to the I&I results, the former performs similarly in terms of estimation quality of the low-frequent component in the wind speed with the advantage of less noisy estimates. Similar to the NREL-5MW results previously, estimation biases are also observed in the IEA-22MW case, which, again, are likely attributed to the inaccuracies in the $C_p$ table of the corresponding turbine and absence of dynamic inflow modeling.

From the aforementioned simulations, aerodynamic power and wind speed estimation error histograms are provided for the considered turbines. Figure 15 depicts the former, where the top row shows the histograms of the NREL-5MW turbine, and



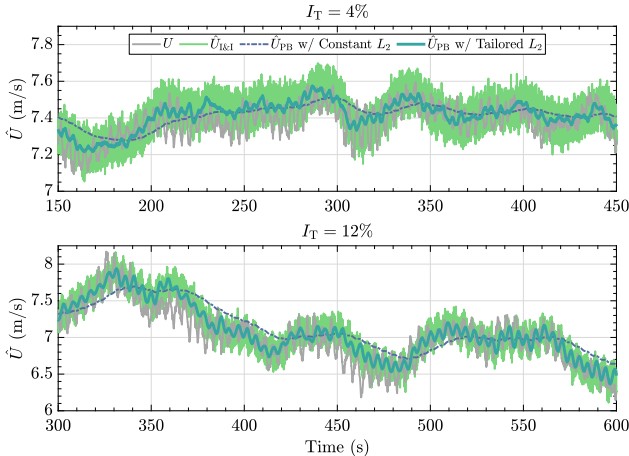

**Figure 14.** Wind speed estimation time series for the IEA-22MW wind turbine. The proposed method gives a smooth, less noisy estimate compared to the I&I (green line) for both turbulent cases. Constant $L_2$ (dashed dark blue line) results in lagged estimation compared to gain-tailored $L_2$ (solid blue line). The actual REWS, as outputted by OpenFAST, is shown in the gray line.

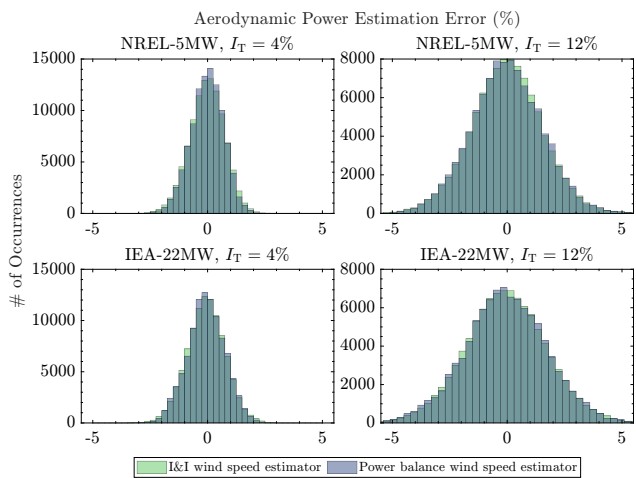

**Figure 15.** Histograms of the wind speed estimators' aerodynamic power estimation errors. Both the I&I and the proposed power balance wind speed estimator are shown to have similar aerodynamic power error distributions.

the bottom row the histograms of the IEA-22MW. For all turbulent cases and estimators, the normalized aerodynamic power estimation errors, defined as $(P_\mathrm{r} - \hat{P}_\mathrm{r})/P_\mathrm{g,rated} \cdot 100\%$, are shown to be similar. Most errors of the I&I and the proposed methods (only shown for the tailored gain case) are concentrated at 0% with decreasing occurrences at larger percentages, resembling a bell curve. At higher turbulence intensity, wider histograms are obtained, which is logical due to the limitation of both estimators in capturing high-frequency contents of the actual REWS.

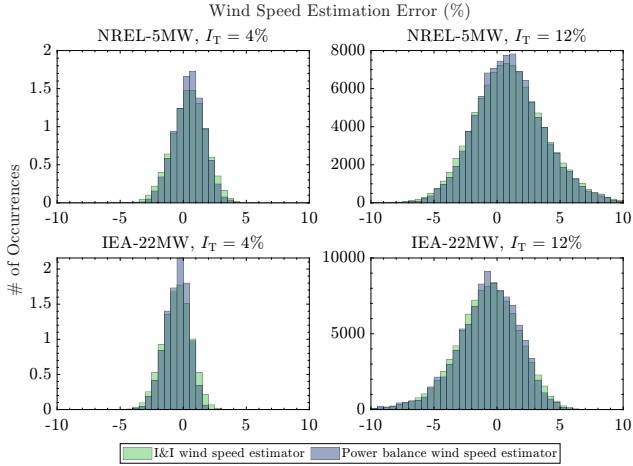

**Figure 16.** Histograms of the wind speed estimators' wind speed estimation errors. Both the I&I and the proposed power balance wind speed estimator are shown to have similar wind speed estimation error distributions. Subtle differences are also shown for both the I&I and the proposed power balance wind speed estimator, where the former has slightly higher occurrences at the histograms' tails due to the higher noise level, while the latter has slightly higher occurrences at the center of the distributions.

Figure 16 shows the histograms of the normalized wind speed estimation errors, defined as $(U - \hat{U})/U_{\mathrm{h}} \cdot 100\%$. The current figure sheds new light on the presence of skewed histograms. For the NREL-5MW turbine, the error histograms are right-skewed, indicating the wind speed tends to be underestimated. On the contrary, the IEA-22MW tends to be left-skewed; that is, the estimators tend to overestimate the wind speed value instead. Given that such skewness is not observed for the aerodynamic power estimation errors, that of the wind speed estimation might come from the inaccuracy of the $C_{\mathrm{p}}$ tables for the corresponding turbines, as previously suspected. Online calibration of such a $C_{\mathrm{p}}$ table has been studied, e.g. in Mulders et al. (2023a, b), which can be used to correct the wind speed estimation quality to minimize such skewness.

Based on the higher-fidelity simulation results reported above, the following conclusions of this section are drawn. First, it has been shown that the mean values of both the power-based wind speed estimator and the I&I are identical. Moreover, it has also been demonstrated that the use of steady-state information, i.e. by $C_{\mathrm{p}}$ tables, in a dynamic environment can lead to skewness for both estimators. Dynamic inflow modeling might also be required for future improvements to the current scheme. Nonetheless, under noisy measurement conditions, the former exhibits more noise resilience, whereas the latter requires additional filtering—complicating the design and potentially introducing phase lag. Finally, the convenience provided by the gain-tailoring for the proposed method has allowed for performance calibration between the small and larger wind turbines.

## 7 Conclusions

In this work, an analysis framework and optimal design for a power balance REWS estimator have been proposed. The estimator is subdivided into two subcomponents based on their role in the scheme, namely, the aerodynamic power estimator



and the wind speed estimate solver. Two aerodynamic power estimator techniques have been thoroughly analyzed, one based on numerical derivative and another based on the Luenberger state estimation technique. Of the two potential aerodynamic
power estimators, the state estimation-based technique has been chosen due to its better resilience against noisy measurements. Moreover, for the first time, a gain-tailoring method for performance calibration throughout a range of modern wind turbine sizes has been formalized. Regarding the wind speed estimate solvers, two options have also been considered, namely the continuous and iterative single-step solvers. In this study, the former's frequency-domain stability analysis has been conducted in the continuous-time domain. Under the Forward Euler discretization, deteriorations in the stability properties of this solver
have been identified and shown in the discrete-time domain. Despite the favorable stability properties for the analyzed Backward Difference and Tustin discretization method, the more robust iterative single-step wind speed estimate solver has been chosen and, in combination with the state estimation-based aerodynamic power estimator, forms the optimal power balance wind speed estimator structure. This optimal power balance wind speed estimator has been validated in the mid-fidelity simulation environment OpenFAST, utilizing the NREL-5MW and IEA-22MW wind turbines, representing small and large wind
turbines in the considered range, respectively. The proposed method and the considered 'baseline' I&I wind speed estimator have shown similar performance in estimating the low-frequent component of the wind speed, with the latter performing good REWS tracking, better noise resilience, and convenient estimator gain calibration across different turbine sizes.

Zero-bias aerodynamic power estimations have been obtained for both estimators; however, time series and histogram analyses have shown the appearance of biased wind speed estimations for both methods. Such biased estimations might be attributed
to the inability of the steady-state $C_\mathrm{p}$ data used to estimate the wind speed in a highly dynamic environment, excluding effects such as dynamic inflow. However, learning algorithms to capture the true power coefficient during operation exist in the literature and can be incorporated to improve the proposed method's performance.

Future study will consider providing an optimal means for the estimator tuning, e.g., by Bayesian optimization and incorporating currently unmodeled dynamics, e.g., drivetrain torsion, tower dynamics, and dynamic inflow effects.

**Appendix A: Improved Immersion and Invariance Wind Speed Estimator**

The improved Immersion and Invariance (I&I) wind speed estimator as studied in Liu et al. (2022) is provided in brief in this section. Readers interested in the detailed derivations and analyses are referred to Liu et al. (2022).

The I&I estimator is described by the following torque balance equation

$$\dot{\hat{\omega}}_\mathrm{r}(t) = \Phi\left(\omega_\mathrm{r}(t), \hat{U}(t)\right) - \frac{T_\mathrm{g}(t)}{J}, \tag{A1}$$

with the following nonlinearity

$$\Phi\left(\omega_\mathrm{r}(t), \hat{U}(t)\right) := \frac{\rho A \hat{U}(t)^3 C_\mathrm{p}\left(\hat{\lambda}(t)\right)}{2 J \omega_\mathrm{r}(t)}. \tag{A2}$$



The wind speed estimate is the result of the minimization of the error between rotor speed and its estimate by a proportional-integral compensator, that is

$$\hat{U}(t) = K_{\mathrm{p}} e_{\omega_{\mathrm{r}} - \hat{\omega}_{\mathrm{r}}}(t) + K_{\mathrm{i}} \int\limits_0^t e_{\omega_{\mathrm{r}} - \hat{\omega}_{\mathrm{r}}}(\tau) \mathrm{d}\tau \,, \tag{A3}$$

with $K_{\mathrm{p}}$ as the proportional gain and $K_{\mathrm{i}}$ the integrator gain, where

$$e_{\omega_{\mathrm{r}} - \hat{\omega}_{\mathrm{r}}}(t) = \omega_{\mathrm{r}}(t) - \hat{\omega}_{\mathrm{r}}(t) \,. \tag{A4}$$

*Author contributions.* AKP: conceptualization, methodology, software, validation, investigation, visualization, writing (original draft). FS: conceptualization, supervision, investigation, writing (review and editing) SPM: conceptualization, methodology, supervision, investigation, writing (review and editing).

*Competing interests.* The authors declare that they have no conflict of interest.

*Acknowledgements.* This project has been performed in cooperation with Vestas Wind Systems A/S.





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
