# Peer review of "Analysis and calibration of optimal power balance rotor-effective wind speed estimation schemes for large-scale wind turbines"

_Wind Energy Science, 2024_

## Author Comment (AC1)

| Date | January 27, 2025 |
| Our reference | n/a |
| Your reference | wes-2024-158 |
| Contact person | Atindriyo K. Pamososuryo |
| Telephone | +31 (0)15 27 85248 |
| E-mail | A.K.Pamososuryo@TUDelft.nl |
| Subject | Author's Response |

**Delft University of Technology**

Delft Center for Systems and Control

Address
Mekelweg 2 (ME building)
2628 CD Delft
The Netherlands

Reviewers
*Wind Energy Science Journal*

Dear reviewers,

The authors would like to thank the reviewers for the constructive and thorough comments and suggestions for our paper. We believe that your feedback has helped us significantly improve the quality of the manuscript.

To consider all the feedback, the paper has been carefully revised. The objective of this document is to reply to the points raised and provide a detailed overview of the changes made. For each comment, a point-to-point response is provided in blue color, while the corresponding changes to the manuscript are reported in red. Please note that, in the enclosed marked-up version of the revised manuscript, the removed and added portions of the manuscript are indicated by red strikethrough text and blue underlined text, respectively. We hope that this document provides satisfying answers to the reviewers' comments.

Yours sincerely,

Atindriyo K. Pamososuryo
Fabio Spagnolo
Sebastiaan P. Mulders

Enclosure(s): General notice
Response to Reviewer 1
Response to Reviewer 2
Response to Reviewer 3
Marked-up version of the revised manuscript

**General notice**

Apart from the changes made in our responses to the reviewers' comments and questions, we include an additional figure for better clarity in the revised manuscript. This new figure (Fig. 5) depicts the internal structure of the state estimation-based aerodynamic power estimator, which is described by the state space system in (18). As a consequence, Figs. 5-16 in the original manuscript become Figs. 6-17 in the revised manuscript. Nevertheless, to prevent confusion in our responses below, we retain the indexing convention of the original manuscript. We thank the reviewers for their understanding.

[Figure]

**Figure 5.** Internal structure of the state estimation-based aerodynamic power estimator. Note that the aerodynamic power estimate $\hat{P}_r$ is the product of the rotor speed estimate $\hat{\omega}_r$ and aerodynamic torque estimate $\hat{T}_r$. See Remark 4 for more details.

**Revised portion prior to (18):**
The general state space expression for the state estimator—as depicted in Fig.5, augmenting (17) to (16) and also including feedback from the measured system output $y = \omega_r$, is written as follows...

**Response to Reviewer 1**

**General Comments**

The paper is well written and makes valuable contributions which opens the path to better and more accurate rotor effective wind estimators. This is critical to achieve effective control and lowering of loads and thus cost on modern very large wind turbines.

The work is based on a number of assumptions which significantly limits the direct applicability of the work. These assumptions are clearly stated, but the expected consequences and limitations that these assumptions entail are not treated. For the paper to give a accurate picture of the contribution made and where further work is needed, I believe that such assumptions should be discussed as part of the paper.

**Response:** Thank you for your kind words and appreciation of our work. We also thank you for your feedback and, in the following, we provide our responses to your specific comments.

**Specific Comments**

1. Assumption 1: No drivetrain loss
   This assumption is obviously not realistic and will easily lead to a rotor power estimation error in the order of 10%. It would be beneficial to describe how this assumption could potential be lifted in order to show the feasibility using the proposed scheme in the presence of uncertain drivetrain losses.

   **Response:** Thank you for your feedback regarding Assumption 1. In this work, first and foremost, our aim was to establish an analysis framework of a power balance wind speed estimation method. This involves segretation between aerodynamic power estimation and wind speed solving submodules and finding the optimal combination given multiple options for each and introduction of a calibration method across a range of turbine sizes. Therefore, to make our derivations and analyses more straightforward and stay aligned with the main goal of our paper, we did not include any drivetrain losses in our original manuscript, thereby the inclusion of Assumption 1.

   That said, we agree that a degree of realism can be further added to the first-principle modeling by including drivetrain losses. To this end, therefore, we include a generator efficiency factor in the revised manuscript. Please note that the inclusion of the generator efficiency has been incorporated in Sections 2-4, including Figs. 4, 5, 6, 11, and 12 of the revised manuscript. We also include an additional explanation in Section 4.1 in this regard.

Related to uncertainties in drivetrain losses, learning algorithms may perhaps be incorporated, for instance, by Gaussian process methods (Hart et al., 2018). This would require modification of the drivetrain model, accommodating uncertainties in the losses. As such additional complexities would require extra work and potentially drift the focus of our current study, we reserve such an extension for future work.

**Revised portions:**

- **(2) and the explanatory text that follows:**
  "...

$$J\omega_{\mathrm{r}}(t)\dot{\omega}_{\mathrm{r}}(t) = P_{\mathrm{r}}(t) - P_{\mathrm{g}}(t)/\underline{\eta_{\mathrm{g}}}\,. \tag{2}$$

  where $J \in \mathbb{R}$ is the low-speed shaft (LSS) equivalent inertia, $\omega_{\mathrm{r}} \in \mathbb{R}$ the rotor angular speed, and $P_{\mathrm{g}} \in \mathbb{R}$ the generated power with the corresponding generator efficiency factor $\eta_{\mathrm{g}} \in (0,1]$. ..."

- **(8) and (14) as a result of the change in (2):**

$$\hat{P}_{\mathrm{r}}(t) = J\omega_{\mathrm{r}}(t)\hat{\dot{\omega}}_{\mathrm{r}}(t) + P_{\mathrm{g}}(t)/\underline{\eta_{\mathrm{g}}}\,. \tag{8}$$

$$\hat{P}_{\mathrm{r}}(k) = J\tilde{\omega}_{\mathrm{r}}(k)\hat{\tilde{\dot{\omega}}}_{\mathrm{r}}(k) + P_{\mathrm{g}}(k)/\underline{\eta_{\mathrm{g}}}\,. \tag{14}$$

- **Third paragraph of Section 4.1.1 (note: including the change of steady-state power resulting from the inclusion of $\eta_{\mathrm{g}} = 0.94$):**
  "...In addition, the inertia value of the NREL-5MW is used (see Table 1), under the steady-state operating conditions $\bar{\omega}_{\mathrm{r}} = 0.8$ rad/s and $\bar{P}_{\mathrm{g}} = \text{1.647}$ $\underline{\bar{P}_{\mathrm{g}} = 1.548}$ MW, the latter of which is computed using (5) with $C_{\mathrm{p}}^{\star} = 0.469$, and $\lambda^{\star} = 6.53$, and $\eta_{\mathrm{g}} = 0.94$. ..."

- **Explanatory text following (15):**
  "...where $G \in \mathbb{R}^{+}$ is the gearbox ratio of the drivetrain, $T_{\mathrm{r}} = P_{\mathrm{r}}/\omega_{\mathrm{r}}$ is the aerodynamic torque, and $T_{\mathrm{g}} = P_{\mathrm{g}}/(\eta_{\mathrm{g}}\omega_{\mathrm{r}}G)$ is the generator torque."

- **Change in the aerodynamic power equation in Remark 4 as a result of the change in (2):**
  "**Remark 4.** Similar to the numerical derivative-based method, the state estimation scheme above can provide the aerodynamic power estimate by making use of the relation $\hat{P}_{\mathrm{r}} = J\hat{\omega}_{\mathrm{r}}\hat{\dot{\omega}}_{\mathrm{r}} + P_{\mathrm{g}}/\underline{\eta_{\mathrm{g}}}$. ..."

- **Last paragraph of Section 4.3:**
  "...as this will diminish and deteriorate the $J\omega_{\mathrm{r}}\dot{\omega}_{\mathrm{r}}$ estimate such that $\hat{P}_{\mathrm{r}} \approx P_{\mathrm{g}}/\underline{\eta_{\mathrm{g}}}$ (i.e. information will be lost). The case where $J\omega_{\mathrm{r}}\dot{\omega}_{\mathrm{r}} = 0$ such that $\hat{P}_{\mathrm{r}} = P_{\mathrm{g}}/\underline{\eta_{\mathrm{g}}}$ is demonstrated by the green line, ..."

- **Section 6.1, second paragraph:**
  "...In contrast to using equal $C_\mathrm{p}$ tables in the previous analysis sections, using the reference wind turbine models in OpenFAST leads to simulating the aerodynamic properties of the respective turbines. For both turbines, $\eta_\mathrm{g} = 1$ is chosen. ..."

2. Assumption 2: Only below rated operation with constant pitch angle

This assumption could potentially make the results of the paper unobtainable in real conditions. The state of the art wind turbine control uses pitch extensively in the below rated region. For e.g. optimal power tracking outside the optimal lambda region, thrust peak shaving, noise reduction control, fore-aft tower damping.

The REWS is also critical for turbine control in the above rated region as various load reduction control techniques rely on accurate wind speed information. Ensuring that the estimator can work in above rated is therefore also of high importance.

From experience designing and tuning wind speed estimators assuming constant pitch angles can lead to poorly tuned wind estimators where the pitch angle signal contaminates the REWS estimate leading to poor dynamic performance. The tuning method procedure derived in this paper might therefore not work if this assumption is lifted.

A discussion on the consequences of lifting this assumption, the feasibility of the presented method when this assumption is lifted and possible mitigations would make the applicability of the method much clearer.

**Response:** Thank you for your remark regarding Assumption 2. We acknowledge the importance of blade pitch angle information to be accounted for in the proposed wind speed estimation framework. In our manuscript, we focus on analyzing the wind speed estimator in open-loop and, therefore, use the baseline $K\omega_\mathrm{r}^2$ controller strategy:

" ...The drivetrain system outputs $\omega_\mathrm{r}$, which is then fed into the optimal torque controller (Bossanyi, 2000), often known as the '$K\omega_\mathrm{r}^2$' controller. However, partial-load controller design is not the main focus of this study; hence, the $K\omega_\mathrm{r}^2$ controller is deemed sufficient for the goal of this work. ..."

Extending the current framework by including an extra input and considering blade pitch angle dependency of the $C_\mathrm{p}$ table inside the wind speed estimator should not be difficult. This approach has also been employed successfully by Lazzerini et al., (2024) and Zahle et al. (2024). For now, we retain the assumption, however, we

acknowledge the valuable suggestion of the reviewer for future formal analysis.

**Revised portions:**

- **Section 3.2, prior to (5):**
  "...That said, this work is equally applicable to more advanced partial-load controllers available in the literature, such as tip-speed ratio tracking schemes, e.g., Brandetti et al. (2023) and Lazzerini et al. (2024). Note must be taken, however, that in the latter scheme, blade pitching is active in partial load. Thus, further study of the current scheme under varying pitch angles is required and reserved for future work. ..."

- **Section 7, last sentence:**
  "...Future study will consider providing an optimal means for the estimator tuning, e.g., by Bayesian optimization, incorporating currently unmodeled dynamics, e.g., drivetrain torsion, tower dynamics, and dynamic inflow effects, and accounting for blade pitch information. "

3. Assumption 3: Accurate power coefficient

Assuming an accurate power coefficient is reasonable given clean blades, however the power coefficient model used in this paper is not suitable for modern very large wind turbine designs such as the IEA reference wind turbines used in this paper. The model neglects the substantial aerodynamic effects of rotor deflection which are especially prevalent in the below rated region.

The effect is not seen in the simulations as the ElastoDyn module is used for the blade beam models and this model does capture the torsion of the blades which is critical for the aerodynamic performance. Instead using the BeamDyn module would show the effect of blade deflection on the estimator performance.

Discussing this aspect would likely give a clear path for future work. The authors could even consider computing the Cp value for a few operating points of the IEA22MW turbine near the rated wind speed with and without blade deflection in order to discuss / assess the magnitude of the error induced

**Response:**
Thank you for your comment and suggestion. This is indeed a very valid point, and we thank the reviewer for raising it. Indeed, present-day large-scale turbines have blades with considerable flexibility under varying loads resulting in bending and twisting largely due to bend-twist coupling effects. In our work, the power coefficient is generated without considering significant deformations in terms of

deflection and structural twist of the blades, which might not produce the most accurate aerodynamic data with respect to real-world conditions, as the reviewer has inferred. Therefore, in our mid-fidelity simulations, we deactivate the Beam-Dyn module to prevent high inaccuracies between the generated power coefficient table and the actual coefficient of the rotor during simulations. Therefore, we have put in our best effort in terms of ensuring consistency between our data generation and simulation conditions.

That said, we acknowledge that having a more representable power coefficient table, followed by evaluation with the higher-fidelity structural module such as BeamDyn, would be beneficial in terms of real-world accuracy. One of the implications of incorporating a blade flexibility model into the aerodynamic power coefficient generation would require a redefinition of the $C_\mathrm{p}$ table as a function of tip-speed ratio $\lambda = \omega_\mathrm{r} R / U$ (where $R$ is the rotor radius) into a function of rotor speed $\omega_\mathrm{r}$ and wind speed $U$. This is because different combinations of $\omega_\mathrm{r}$ and $U$ would not give the same power coefficient due to changing aerodynamic properties as a result of blade bending and bend-twist coupling even though they produce the same $\lambda$. The previously mentioned approach involves extended steady-state calculations of the aerodynamic properties in steady state resulting from the added dimension and has been recently exposed and elaborately analyzed in the work of Lazzerini et al. (2024).

As discussed above, considerations in accounting for significant blade deformation in generating a power coefficient table will add complexity to our current analysis presented in the paper, for instance by replacing $C_\mathrm{p}(\lambda)$ into $C_\mathrm{p}(\omega_\mathrm{r}, U)$. Aligned with our earlier response on Assumption 1, our primary goal is to develop a framework for analyzing a power balance wind speed estimation method, identifying its optimal structure among subcomponent combinations, and calibrating it for different turbine sizes. Hence, we must limit the scope of our current analysis, and therefore, alterations in the behavior and performance of the proposed estimator under more flexible blades are subject to future study. Nevertheless, to enhance the clarity of our current work, we include a new remark in the revised manuscript as documented below.

**Revised portion, following (4):**
"**Remark 1**. The power coefficient considered in this work does not take into account the aerodynamic effects due to structural deformations, e.g., those associated with bend-twist coupling of the blades. Had this been the case, changes in local blade sections' angle of attack are expected and different combinations of $\omega_\mathrm{r}$ and $U$, although they correspond to the same $\lambda$, might yield different power coefficients. This would render the one-on-one mapping between $\lambda$ and $C_\mathrm{p}$ inadequate such that $C_\mathrm{p} : \mathbb{R}^2 \to \mathbb{R}$ mapping is needed (i.e., $C_\mathrm{p}(\omega_\mathrm{r}, U)$ instead of $C_\mathrm{p}(\lambda)$; see Lazerrini, et al. (2024) and references therein). Nevertheless, without loss

of generality, the mapping of the former is adopted for the sake of clarity of the analysis of this paper; thus, the $C_\mathrm{p}$ tables in this work are generated using rigid rotor assumption. "

**Technical Corrections**

1. Seems that the Fast simulations are sometimes referred to as mid-fidelity, sometimes high-fidelity. e.g. line 85 and 95.

   **Response:** Thank you for your comment. Throughout our paper, we refer to OpenFAST simulations as mid-fidelity, especially in the introduction and conclusion chapters. At times, we also refer to it as higher-fidelity to contrast it with the lower-fidelity simulations in Sections 4-5. Nevertheless, to prevent confusion, in the revised manuscript, we consistently use 'mid-fidelity.' Please see the attached marked-up version of the revised manuscript for details

2. Line 116 It is claimed that the reference turbines represent industrial turbine designs which is not accurate since these turbines are designed by academia. If you compare the relations of the listed quantities across industrially designed turbines I expect you will find a high variance.

   **Response:** Thank you for raising this point. It is indeed true that the presented reference wind turbines are designed by academia. Moreover, we must acknowledge that whether the listed reference wind turbines in our original manuscript are representative of industrial turbine designs still requires more rigorous data collection from various original manufacturers and further study. Therefore, we remove the following statement in the revised manuscript.

   **Revised portion in Section 2.2, at the beginning and the end of the section, respectively:**
   "As mentioned earlier in Section 1, wind speed estimator calibration  methodology for various wind turbine sizes is presented in this work.
   . . .
   "

3. Line 240 Often the "noise" seen on the speed measurement are unmodelled dynamics and not Gaussian white noise. Some of these unmodelled dynamics will be present in the fast simulation. The filtering chosen thus often needs to consider the unmodelled dynamics rather than the actual noise variance

**Response:** Thank you for your remark. We acknowledge that, in real wind turbine speed measurements, additional dynamic effects than what is modeled in our work can be observed and may deteriorate wind speed estimation if not fully accounted for.

We agree that including additional dynamics, such as additional drivetrain DOF, tower DOFs, unsteady aerodynamics, and any residual unmodeled dynamics such as noise, may further improve the wind speed estimation quality. As incorporating such dynamics in the current work would potentially deviate from its main focus, we intend to do this in the future, as we have already inferred in the revision below.

**Revised portion, last paragraph of Section 7:**
" . . . Future study will consider providing an optimal means for the estimator tuning, e.g., by Bayesian optimization and incorporating currently unmodeled dynamics, e.g., drivetrain torsion, tower dynamics, and dynamic inflow effects. "

4. Line 340 The turbulence intensity used for the simulations is very low. What justified using such low values?

**Response:** Thank you for your question. Although we present our work as a site-independent framework, we believe that 4% of turbulence intensity is a reasonable choice, e.g., for offshore sites (see, e.g., Van der Hoek et al. (2020) and Taschner, E. et al. (2023)). Furthermore, despite the use of low turbulence presented in the manuscript, we also performed simulations with higher turbulent wind cases, i.e., 12% to prove the efficacy and validity of our method. This number is a reasonable choice for onshore sites, as shown in the following figure.

[Figure]

The above results show high similarities with respect to that of the 4% turbulence as shown in Fig. 5 of the original manuscript, despite the change in overall magnitudes. As we reach the same conclusions with the low turbulent case, we do not include the high-turbulent results in the paper.

5. Figure 5: It would be beneficial to include the normalized power error and maybe even the cubic root of the normalized error as this should be proportional to the resulting error in estimated wind speed.

**Response:** Thank you for your comment. In the following figure, we plot the normalized aerodynamic power error and the corresponding wind speed estimate error, where tailored $L_2$ gain setting is used and iterative single-step wind speed estimate solver is employed for the latter. On the left plot, we show the mean values and on the right plot, we depict the standard deviation values of both errors. The data (and the fitted lines) confirm your statement regarding the proportionality relationship between both errors in terms of mean and standard deviation.

[Figure]

While this finding confirms your statement, we believe it serves as a supplementary validation to the core contributions of our work. Therefore we refrain from its inclusion in the revised manuscript.

6. Line 574 As part of assessing the performance of the estimator it would be beneficial to have step wind or gust simulation cases where it is easy to see the transient response e.g. lag. Minimizing the lag is very important for the turbine control in gust like scenarios.

   **Response:** We appreciate the suggestion to demonstrate transient response in step wind or gust scenarios. However, transient characteristics are part of the dynamic tuning of the estimator, which would change for different tunings. Thus, we heavily focus on providing tuning rules and methodology for satisfactory estimator performance over a turbine range; the actual dynamic transient behavior is a design choice of the one implementing and calibrating the estimator.

7. Figure 13: A zoom would nice as the both the actual REWS and the I&I REWS have higher frequency content and it is not possible to see if they are actually correlated

   **Response:** Thank you for your suggestion. We agree with you that zoomed-in insets would enhance the clarity of the plots, especially in providing information on whether the actual REWS and the I&I estimate are correlated. Therefore, in the revised manuscript, we changed Figs. 13 and 14 into the following to include zoomed-in plots. As can be seen in the 'Revised portions' below, some oscillations of the I&I REWS estimation seem to be correlated with that of the actual REWS with some lags. However, without additional tools, it is challenging to present any clear conclusions regarding the correlations of both signals. Therefore we do not do so in the current work to retain its conciseness and clarity.

   **Revised portions:**

[Figure]

**Figure 13.** Wind speed estimation time series for the NREL-5MW wind turbine. The proposed method (blue line) gives a smooth, less noisy estimate compared to the I&I (green line) for  low (top plots) and high (bottom plots) turbulent cases. The low-frequent component of the wind speed is captured where biased estimation occasionally occurs, potentially due to an inaccurate $C_{\mathrm{p}}$ table utilized in both estimators and an absence of dynamic inflow modeling in the estimators. The actual REWS, as outputted by OpenFAST, is shown in the gray line. Zoomed-in plots of 25-s time spans are provided on the right for clearer observation of the estimation performance.

[Figure]

**Figure 14.** Wind speed estimation time series for the IEA-22MW wind turbine. The proposed method gives a smooth, less noisy estimate compared to the I&I (green line) for  low (top plots) and high (bottom plots) turbulent cases. Constant $L_2$ (dashed dark blue line) results in lagged estimation compared to gain-tailored $L_2$ (solid blue line). The actual REWS, as outputted by OpenFAST, is shown in the gray line. Zoomed-in plots of 25-s time spans are provided on the right for clearer observation of the estimation performance.

8. Figure 15: Very hard to read as the bars blue bars cover the green bars. Consider

changing from bars to lines, such that both distributions can easily be seen,

**Response:** Thank you for pointing this out. We agree that Fig(s) 15 (and 16) can be further improved by changing the presentation from bars to lines. Therefore the following figure has been used to replace Figures 15 (and 16).

**Revised portion:**

[Figure]

**Figure 15.** Aerodynamic power estimation error histograms of the wind speed estimators. Both the I&I and the proposed power balance wind speed estimator are shown to have similar aerodynamic power error distributions.

[Figure]

**Figure 16.** Wind speed estimation error histograms of the wind speed estimators. Both the I&I and the proposed power balance wind speed estimator are shown to have similar wind speed

estimation error distributions. Subtle differences are also shown for both the I&I and the proposed power balance wind speed estimator, where the former has slightly higher occurrences at the histograms' tails due to the higher noise level, while the latter has slightly higher occurrences at the center of the distributions.


**Response to Reviewer 2**

The paper discusses the topic of estimating the wind speed for wind turbines, the topic is relevant to wind energy industry. The paper is clear and well written, and it add a contribution to the field of wind turbine wind speed estimation, the methods used to develop both the aerodynamic power estimator and wind speed estimate solver has been well implemented and described.

**Response:** Thank you for the kind words and appreciation given to our work. In the following we provide our responses to your comments.

Nevertheless, the assumptions of neglecting the power losses between the generator and the gearbox of the wind make the method developed not suitable for direct implementation, the efficiency of the drivetrain can be added to the problem in future study based on the outcomes of this work.

**Response:** Thank you for your comment. In the original manuscript, we did not account for the losses between the generator and the gearbox of the wind turbine. However, in the revised manuscript, we have addressed this issue by lifting the assumptions, specifically Assumption 1, as detailed in our response to Reviewer 1's Specific Comment No. 1. We kindly refer you to that response for further clarification.

There are minor typos observed in the manuscript as follow: 1- Page 3 line 76 "...outlines the contributions of this work" a colon (:) could be added after 'work'. 2- Page 7 line 157, the expression at the end of the line exceeds the page margin. 3- The front size in figures [1, 4 - 6, 8, 9, 11 - 16] is small and could be enlarged.

**Response:** Thank you for pointing these out. In the revised manuscript, we have applied changes to improve the readability of our paper, including: (1) adding a colon on page 3 line 76, (2) fixing the page margin exceeded by the expression at the end of line 157 of page 7, and (3) increasing the font size in figures [1, 4 - 6, 8 - 16]. Please also note that we altered the width vs. height ratio of Figs. 6, 11, and 12 in the revised manuscript to reduce whitespace while still maintaining clarity.

**Response to Reviewer 3**

The paper addresses the topic of wind estimation, which is of great relevance in wind turbine engineering, and uses a systematic approach and well-known tools such as Kalman filters and Luenberger estimators. Besides, the method has been validated using aeroelastic simulations. However, some important issues should be addressed before the paper is published:

1. The introduction states that many wind turbine control structures are based on wind estimators. However, this is far from true. Wind turbine controllers are mainly based on the feedback of generator speed and nacelle acceleration because wind measurements and estimates should not be trusted or have to great a delay. How does the proposed method change this scenario?

   **Response:** Thank you for your question. While it is true that many wind turbine controllers rely heavily on feedback from generator speed and nacelle acceleration, the statement in the introduction has already provided evidence referring to the growing body of research and emerging wind turbine control approaches incorporating wind speed estimators. These methods, though they might not be universally adopted, are gaining traction due to advancements in estimation techniques and their potential to improve performance in certain scenarios. Further evidence has also been found in recent publications, which we have included in the revised manuscript:

   (a) In the work of Brandetti et al. (2023), the authors provide a rigorous study on the comparison between wind speed estimate-tip-speed ratio (WSE-TSR) tracking scheme and a baseline $K\omega_\mathrm{r}^2$ controller. It is shown that the WSE-TSR scheme provides more flexibility compared to the baseline controller (which solely relies on generator speed measurements) in terms of power and load objectives trade-off;

   (b) In the work of Lazzerini et al. (2024), the authors propose a novel feedforward-feedback control scheme that leverages optimal operational set points computed by an optimizer accounting for the effects of blade deformations on the aerodynamic performance and turbine loading. In their proposed scheme, the authors utilize wind speed estimates to schedule their proposed controller to track the generated optimal set points, as well as to schedule a set point smoother for a smooth partial-full load transition. This work has been established in collaboration with an industry partner;

   (c) The Reference Offshore Open-Source Controller (ROSCO) (Abbas et al., 2022)—developed to represent industrial standards—provides wind speed estimation as a means to provide wind speed information to a tip-speed ratio tracker torque controller, not to mention a pitch saturation routine.

(d) The power balance wind speed estimation scheme that we proposed has been utilized in previous studies in Mulders et al. (2023a) and Mulders et al. (2023b), where the combination of numerical derivative-based aerodynamic power estimator and continuous wind speed estimate solver is chosen as the structure of the estimator. Therein, the wind speed estimate is leveraged in a WSE-TSR tracking scheme to provide the optimal reference for tracking the optimal TSR at the partial-load regime. In these studies, however, the authors focus on $C_{\mathrm{p}}$ learning algorithms development and not on the analysis of the power balance wind speed estimator itself. These works (and our current work) have also been established in collaboration with an industry partner.

With regard to wind speed estimation delays/lags, Section 4 of our study has been dedicated to the analysis of such undesired behavior. Particularly, we have identified that information lack due to excessive noisy measurement filtering in the numerical derivative-based aerodynamic power estimation would lead to $\hat{P}_{\mathrm{r}} \approx P_{\mathrm{g}}/\eta_{\mathrm{g}}$, which is the worst-case lag scenario as presented in Fig. 6. As such a lag would entail a lagged wind speed estimate, we propose the state estimation-based aerodynamic power estimator as a viable alternative. This is because the estimation performance can be tuned intuitively by choosing appropriate natural frequency and damping ($\omega_0$ and $\zeta_0$) as elaborated in Section 4.3 such that minimum delays/lags can be obtained while achieving overall high estimation accuracy. The resulting wind speed estimate thus becomes a reliable and valuable signal for augmentation with controllers (the cases of which have been demonstrated in the aforementioned literature, as well as in Section 1 of our manuscript).

We hope this clarifies our perspective.

**Revised portion, second paragraph of Section 1:**
"Regardless of the potential economic benefit, the task of controlling wind turbines with larger rotors is becoming more of a challenge, especially when accurate information on wind speed is crucial to ensure high controller performance, e.g., for gain-scheduling (Kumar and Stol, 2009; Koerber and King, 2013), feedforward(-feedback) control (Van Engelen and Van der Hooft, 2003; Koerber and King, 2013; Lazzerini et al., 2024), or tip-speed ratio tracking (Bossanyi, 2000; Ortega et al., 2013; Abbas et al., 2022; Brandetti et al., 2022, 2023), to mention a few...."

2. What is the main purpose of the proposed wind estimate? If it is going to be used for control purposes, how does the estimator block improve performance? As the control action is going to be based on the generator speed either way, why is a two step procedure (estimator+controller) a better option than the standard feedback? How does the main purpose affect the requirements on the estimated signal?

**Response:**

Thank you for your questions. One of the main purposes of the proposed wind speed estimate is indeed for control purposes, as you have indicated. Nonetheless, in our work, the used controller $(K\omega_\mathrm{r}^3)$ is solely reliant on the generator speed and does not use any wind speed information, as you have also inferred. This is done purposefully to allow as clear as possible analysis of our proposed estimator in an open loop—free from controller interferences, similar to what was done in the literature, e.g., Østergaard et al., (2007) and Soltani et al., (2013). That being said, we, in contrast to the reviewer's observation, see a large industrial traction for advanced controllers including the wind speed estimator and tip-speed ratio tracking (WSE-TSR) schemes. The use of such a controller is illustrated by the works mentioned above. Nevertheless, several aspects of the estimator remained unclear, which we aimed to address in our current work, such as

(a) how the *estimated signal* would look like under noisy measurements;

(b) how the estimator should be calibrated across different turbines such that the *estimated signal* is representative with respect to the sizes of the turbines in question;

(c) whether or not there are better alternatives (thus more optimal estimator structure and better-conditioned resulting *estimated signal*) than the existing aerodynamic power estimator and wind speed estimate solver.

The answer to the reviewer's question on whether a two-step procedure is a better option than the standard feedback might lie in the study of Brandetti et al. (2023). In the study, the authors investigated a WSE-TSR tracking procedure akin to that used in Mulders et al. (2023a) and Mulders et al. (2023b) and made a comparison with a standard $K\omega_\mathrm{r}^2$ controller—with the Immersion and Invariance wind speed estimator being used in place of the power balance estimator. The authors demonstrate in that work that by having such a combined estimator-controller scheme, more freedom in dynamic controller tuning is attained. Moreover, there are greater possibilities in reaching optimal actuation and performance trade-offs, for example limiting generator torque fluctuation and maximizing power production, these optimal solutions are illustrated by Pareto fronts by virtue of additional tuning parameters the framework provides.

3. The estimator is based on the idea that both the inertia and the cP are known values. Is that a realistic approach? Have the authors performed a sensitivity analysis?

**Response:** Thank you for your questions. When a turbine is newly manufactured, the modeled inertia $J$ and power coefficient $C_\mathrm{p}$ are believed to be good estimates of the actual properties of the turbine. However, these properties may change over the lifetime of the turbine due to various causes of degradation. A change of $J$, according to our framework, would result in a change in the dynamics of the state estimation-based aerodynamic power estimator, in particular, the estimator gain $L_2$ would no longer be accurate as it is parameterized by $J$, as shown in (22b). Having an estimate of $J$ from field tests, for instance, by deceleration test (Paniagua and Yasa, 2007; Ilina, 2011; and Jager, 2017), one may retune $L_2$ based on (22b).

With regard to $C_\mathrm{p}$ value learning, the work of Mulders et al. (2023a) and Mulders et al. (2023b), for instance, have provided several learning algorithms for $C_\mathrm{p}$ table calibration given the inaccuracies that might occur in real-world setups. The former of which proposes a wind speed measurement-free approach, whereas the latter proposes an excitation-free and data-driven $C_\mathrm{p}$ calibration approach. Moreover, in the work of Brandetti et al. (2022), sensitivity analysis on the effect of inaccurate $C_\mathrm{p}$ table on wind speed estimation has been studied, where biased estimates might result from the inaccuracy of the coefficient. Estimating such quantities, however, falls within the scope of the parameter estimation and learning control field and is not the focus of our current work; therefore, the reviewer is referred to the aforementioned works for more details.

4. The paper says that high power wind turbines show great wind speed variance across the rotor. How does it affect the estimation? Would it make sense to focus on blade effective wind speed?

    **Response:** Thank you for your question. The greater the rotor area of a wind turbine, the greater the wind speed variation across the rotor. The whole concept of rotor-effective wind speed (REWS) is, therefore, to capture this effect in a single quantity—capturing the spatial variation effects (e.g., from shear, veer, etc.).

    Blade-effective wind speed (BEWS) is another means to capture such spatial variance, for instance, by exploiting blade load measurements and the so-called cone coefficient, the detail of which is provided in the literature, such as the work of Liu et al. (2022). However, whether BEWS is required needs further context and justification as it is needed mostly for more sophisticated control objectives such as blade load mitigation by individual pitch control (e.g., as done in Pamososuryo et al., (2023)), whereas REWS is already sufficient for most control purposes, such as tip-speed ratio tracker torque controller (Brandetti et al., (2023) and Lazzerini et al., (2024)), and blade pitch saturation (Abbas et al., 2022), to mention a few. Based on this reasoning, we focus on the development of the proposed REWS estimator.

5. Section 4.1 proposes a first order filter to eliminate measurement noise. Could the time constant be chosen by characterizing the spectrum of the sensor noise and the generator speed? Maybe a frequency domain analysis could be a useful guide.

**Response:** Thank you for your question. It is indeed possible to choose the time constant by characterizing the spectrum of the sensor noise and the generator speed and then setting the cut-off frequency of the filter accordingly. In Section 4.1 we have provided a measurement noise-time constant sweep, in which we can observe the trade-off between the noise filtering and biased estimate. Later on in Section 4.3, we compare this method with the state estimation-based approach, in which the latter is shown to be much more effective and noise-resilient. Therefore, we do not include additional studies of the filtered derivative parameter characterization in our work, as it is of less interest.

6. The grammar should be reviewed, specifically the use of the Saxon genitive for nonanimate subjects and the subject-verb agreement.

**Response:** Thank you for your feedback. In the revised manuscript, we have made improvements wherever possible with regard to the misuse of the Saxon genitive in the previous version of our work. Subject-verb agreements have also been improved in the revised manuscript. Please refer to our marked-up version of the revised manuscript to see the detailed changes.

Liu, Y., Pamososuryo, A. K., Mulders, S. P., Ferrari, R. M. G., and van Wingerden, J.W.: The Proportional Integral Notch and Coleman Blade Effective Wind Speed Estimators and Their Similarities, 
[revised manuscript text omitted]

---

## Referee Report (RR1)

**1 General comments**

This paper is very well written and clearly sets out its assumptions and limitations. Remarks are well inserted in the text and allow the reader to follow the line of thought and to keep in mind important points. All of this make the comprehension much easier and was very appreciated.

Good wind speed estimation is a hot topic with the integration of such data in control strategy (TSR-tracking, feedforward-feedback, pitch saturation etc...) as mentioned by the authors in the responses to reviewers for revision 2.

The use of mid-fidelity simulations provide useful insights on the use of tuning made with lower fidelity assumptions.

I did not check all the mathematical developments in this paper and will focus on the rest.

**2 Specific comments**

- 1. Line 260: the generator efficiency of the NREL5MW is taken as  $\eta_g = 0.94$  for some simulations. However, in section 6.1, line 560, generator efficiency is taken as  $\eta_g = 1$  for both turbines. Did you use an efficiency of  $\eta_g = 1$  when generating the power coefficient Cp tables? Moreover, is there a specific reason not to use the generator efficiency values provided in the models?
- 2. Immersion and Invariance method for wind speed estimation was mentioned. In the well-known ROSCO controller, Extended Kalman Filter is also implemented. Did you compare against it or do you have an opinion about it?
- 3. It would be very interesting to see the effects of various wind speed estimators on the power production itself if it is used in TSR-tracking control strategies for example.
- 4. While it was already addressed in previous responses, I would like to reiterate on the Assumption 1 comment. Pitch control before the rated power is indeed very useful. The methodology presented in this paper could however be adapted to multi-entries power coefficient Cp mappings, which is satisfying.
- 5. Once more, it was already addressed in previous responses, but using ElastoDyn module that does not account for blade torsion has shortcomings, especially for large rotor such as the IEA22MW. Will you try to include multi-entries Cp mappings in future works?

**3 Technical corrections**

- 1. Lines 154-156. I understand why the  $K\omega_r^2$  is used but I think one sentence is missing to explain the limitations of such a controller. As it is not straightly presented, it can be a bit confusing to follow why there is a point beginning with "However" just after that.
- 2. While on screen, it is not too hard to distinguish the curves, the choice of color scheme (blue, green, gray) make it a bit hard to see clearly the differences on Figures 14 and 15 when the article is printed. It was especially true in my case for the zoom on bottom-right of Figure 14 where I had to refer to the original pdf to better see the bias.

---

## Author Response (AR2)

| | |
|---|---|
| Date | March 6, 2025 |
| Our reference | n/a |
| Your reference | wes-2024-158 |
| Contact person | Atindriyo K. Pamososuryo |
| Telephone | +31 (0)15 27 85248 |
| E-mail | A.K.Pamososuryo@TUDelft.nl |
| Subject | Author's Response |

**Delft University of Technology**

Delft Center for Systems and Control

Address
Mekelweg 2 (ME building)
2628 CD Delft
The Netherlands

Reviewers
*Wind Energy Science Journal*

Dear reviewers,

The authors would like to thank the reviewers again for their constructive and thorough comments and suggestions for our paper. We believe that your feedback has helped us significantly improve the quality of the manuscript. To take into account all the feedback, the paper has been carefully revised. One of the reviewers provided additional comments. This document aims to further clarify and enhance our paper, ensuring all concerns are addressed and eliminating any remaining doubts for publication. For each comment, a point-to-point response is provided in blue color, while the corresponding changes to the manuscript are reported in red. Please note that, in the enclosed marked-up version of the revised manuscript, the removed and added portions of the manuscript are indicated by red strikethrough text and blue underlined text, respectively. We hope that this document provides satisfactory responses to reviewer 1's comments.

Yours sincerely,

Atindriyo K. Pamososuryo
Fabio Spagnolo
Sebastiaan P. Mulders

Enclosure(s): Response to Reviewer 1
Marked-up version of the revised manuscript

**Response to Reviewer 1**

**General Comments**

This paper is very well written and clearly sets out its assumptions and limitations. Remarks are well inserted in the text and allow the reader to follow the line of thought and to keep in mind important points. All of this make the comprehension much easier and was very appreciated.

Good wind speed estimation is a hot topic with the integration of such data in control strategy (TSRtracking, feedforward-feedback, pitch saturation etc...) as mentioned by the authors in the responses to reviewers for revision 2.

The use of mid-fidelity simulations provide useful insights on the use of tuning made with lower fidelity assumptions.

I did not check all the mathematical developments in this paper and will focus on the rest.

**Response:** Thank you for your kind words and appreciation of our work. We also thank you for your feedback, and in the following, we provide our responses to your specific comments and technical corrections.

**Specific Comments**

1. Line 260: the generator efficiency of the NREL5MW is taken as $\eta_\mathrm{g} = 0.94$ for some simulations. However, in section 6.1, line 560, generator efficiency is taken as $\eta_\mathrm{g} = 1$ for both turbines. Did you use an efficiency of $\eta_\mathrm{g} = 1$ when generating the power coefficient $C_\mathrm{p}$ tables? Moreover, is there a specific reason not to use the generator efficiency values provided in the models?

   **Response:** Thank you for your question. Indeed, we used $\eta_\mathrm{g} = 1$ when generating the power coefficient $C_\mathrm{p}$ tables. In fact, in our toolset, the choice of $\eta_\mathrm{g}$ does not affect the recorded $C_\mathrm{p}$ data as this quantity is on the rotor side of the drivetrain. Thus, using the actual generator efficiency values in the mid-fidelity simulations would not change the results and conclusions already provided in our manuscript. For such a reason (and also for the sake of simplicity), we employed $\eta_\mathrm{g} = 1$ in the original manuscript. That said, in the revised manuscript, we use the original efficiency factors for both turbines. Please note that we also rerun the simulations and the provided figures in Section 6 of the revised manuscript have already included the corresponding results, which are identical to the previous version of the manuscript.

**Revised portions:**
" The generator efficiency factors for both turbines are $\eta_{\mathrm{g}} = 0.94$ for the NREL-5MW and $\eta_{\mathrm{g}} = 0.954$ for the IEA-22MW."

2. Immersion and Invariance method for wind speed estimation was mentioned. In the well-known ROSCO controller, Extended Kalman Filter is also implemented. Did you compare against it or do you have an opinion about it?

   **Response:** Thank you for your question. As you mentioned, an Extended Kalman Filter (EKF) in the ROSCO controller is also provided as an alternative to the Immersion and Invariance (I&I) wind speed estimator. In our opinion, the EKF has the least resemblance to our proposed estimator compared to the latter. For instance, its design is more emphasized on the explicit internal wind speed modeling of the wind speed estimator, such as the separation of slow- and fast-varying components in the wind. As a consequence, the tuning guidelines of the EKF rely more on wind characteristics, such as turbulence length scale and turbulence intensity, which may not be made known a priori (although a conservative approach may still be made by choosing hardcoded values as exemplified in Abbas et al. (2022)). Moreover, being a subset of the Kalman filter, the estimation by EKF is done in stages, being the prediction and measurement updates, not to mention the necessity to obtain the Jacobians of the internal model every time step.

   The I&I wind speed estimator and our method, on the other hand, do not rely on dynamic modeling of the wind, as well as its characteristics, for the tuning of the estimators. Furthermore, implementation-wise, these frameworks are at a lower level of complexity compared to the EKF in that they do not require the Jacobians of the internal model to be derived every time step, nor do they involve prediction and measurement updates. On top of that, we also investigated and made enhancements to I&I in terms of added integrator term, thus providing an internal estimated rotor speed state (Liu et al., 2022). Therefore, based on these reasonings and similarities of the I&I estimators to our proposed WSE, we used them as cases for comparison in the current work.

3. It would be very interesting to see the effects of various wind speed estimators on the power production itself if it is used in TSR-tracking control strategies for example.

   **Response:** Thank you for your comment. We agree with you that the effects of various wind speed estimators on power production if they are integrated into TSR-tracking control strategies (such as done in Brandetti et al. (2023)), would be an interesting topic to investigate. Nevertheless, evaluation of the wind speed

estimators in a closed-loop within a TSR-tracking control strategy would increase the complexity and deviation from the main scope of our study. For such reasons, this is only possible to do in future work.

4. While it was already addressed in previous responses, I would like to reiterate on the Assumption 1 comment. Pitch control before the rated power is indeed very useful. The methodology presented in this paper could however be adapted to multi-entries power coefficient Cp mappings, which is satisfying.

**Response:** Thank you for your reiteration of the comment on Assumption 1. Pitch control before rated power is indeed a very relevant varying quantity in partial load control in modern wind turbine controllers, especially for large flexible wind turbines, to achieve various goals (see Lazzerini et al. (2024)). Indeed, the adaptation of our methodology to multi-entries power coefficient $C_{\mathrm{p}}$ mapping increases the WSE performance (consistency of REWS estimation) by making it a function of the current pitch angle, as well. Therefore, we removed Assumption 1 and included $\beta$ dependency of $C_{\mathrm{p}}$ in the current manuscript; it would, however, be an interesting research topic to investigate its effects in future publications/research.

**Revised portions in (3) and text that follows:**
"The aerodynamic power is given by the nonlinear relation

$$P_{\mathrm{r}}(t) = \frac{1}{2}\rho A_{\mathrm{r}} C_{\mathrm{p}}(\lambda(t), \underline{\beta(t)})U(t)^3 \,, \tag{3}$$

in which $\rho \in \mathbb{R}$, denotes the air density, $A_{\mathrm{r}} \in \mathbb{R}$ the rotor area,  $U \in \mathbb{R}$ the REWS (Soltani et al., 2013), and $\beta \in \mathbb{R}$ blade pitch angle. The power coefficient  $C_{\mathrm{p}} : \mathbb{R}^2 \to \mathbb{R}$ is a nonlinear mapping from $\underline{\beta}$ and the non-dimensional tip-speed ratio (TSR), defined as . . .

**Remark 1.** . . . This would render the  $C_{\mathrm{p}} : \mathbb{R}^2 \to \mathbb{R}$ mapping between $\lambda$, $\underline{\beta}$, and $C_{\mathrm{p}}$ inadequate such that  $C_{\mathrm{p}} : \mathbb{R}^3 \to \mathbb{R}$ mapping is needed (i.e.,  $C_{\mathrm{p}}(\omega_{\mathrm{r}}, U, \beta)$ instead of $C_{\mathrm{p}}(\lambda, \beta)$; see Lazzerini et al. (2024) and references therein). . . .

. . .  Although controllers performing better than $K\omega_{\mathrm{r}}^2$, e.g., during transients, are available in the literature, partial-load controller design is not the main focus of this study. Hence, the $K\omega_{\mathrm{r}}^2$ controller is deemed sufficient for the goal of this work. . . .  As standard $K\omega_{\mathrm{r}}^2$ scheme does not utilize blade pitch control, further study of the current estimation scheme under varying pitch angles is required and reserved for future work. Furthermore, constant pitch at fine position $\beta = \beta^\star$ is used and, for the sake of brevity, the notation $C_{\mathrm{p}}(\lambda) := C_{\mathrm{p}}(\lambda, \beta^\star)$ is made use of for the remainder of this paper. . . ."

5. Once more, it was already addressed in previous responses, but using ElastoDyn module that does not account for blade torsion has shortcomings, especially for large rotor such as the IEA22MW. Will you try to include multi-entries Cp mappings in future works?

**Response:** Thank you for your question. We agree that the ElastoDyn has limitations in terms of simulating blade torsion, in particular for turbines with large rotors such that the use of the higher-fidelity blade dynamics module BeamDyn would have been required. As such, multi-entries $C_\mathrm{p}$ tables would have been needed, as you have indicated. Nevertheless, as the inclusion of such an additional module would require additional efforts for the current study, we can only do so in future work. In alignment with our previous response to your Specific Comment 3 (Assumption 3: Accurate power coefficient), we include an additional description in the first paragraph of Section 6 as follows for better clarity of our manuscript.

**Revised portions:**
"...For the validation purposes of this work, the AeroDyn, ServoDyn, ElastoDyn, and InflowWind modules of OpenFAST are used. The BeamDyn module, capable of simulating blade structural dynamics including blade torsion and bend-twist coupling, is not considered in this work; otherwise, higher-dimensional coefficient tables would have been necessary, as discussed in Lazzerini et al. (2024) (see also Remark 1). Future work shall include BeamDyn module usage, where higher-dimensional coefficient tables are required for validations.

Concerning the degrees of freedom (DOFs) of the simulated wind turbines, the following are activated: ..."

**Technical corrections**

1. Lines 154-156. I understand why the $K\omega_\mathrm{r}^2$ is used but I think one sentence is missing to explain the limitations of such a controller. As it is not straightly presented, it can be a bit confusing to follow why there is a point beginning with "However" just after that.

**Response:** Thank you for your feedback with regard to a missing explanation on the limitation of the $K\omega_\mathrm{r}^2$ controller. Therefore, in the revised manuscript, we include the following sentence to enhance further the coherence of the text.

**Revised portions, following Remark 1:**
"The drivetrain system outputs $\omega_\mathrm{r}$, which is then fed into the optimal torque controller (Bossanyi, 2000), often known as the '$K\omega_\mathrm{r}^2$' controller.  Although controllers performing better than $K\omega_{\mathrm{r}}^2$, e.g., during transients, are available in the literature,  partial-load controller design is not the main focus of this study . Hence, the $K\omega_{\mathrm{r}}^2$ controller is deemed sufficient for the goal of this work. That said, this work is equally applicable to more advanced partial-load controllers available in the literature, such as tip-speed ratio tracking schemes, e.g., Brandetti et al. (2023) and Lazzerini et al. (2024). . . ."

2. While on screen, it is not too hard to distinguish the curves, the choice of color scheme (blue, green, gray) make it a bit hard to see clearly the differences on Figures 14 and 15 when the article is printed. It was especially true in my case for the zoom on bottom-right of Figure 14 where I had to refer to the original pdf to better see the bias.

**Response:** Thank you for pointing this out. We agree that the color scheme chosen for Figs. 14-15 makes our results hard to distinguish. Therefore, in the revised manuscript, we adjusted the color scheme so that the results from different quantities are more distinguishable, both on-screen and printed, as well as color-blind friendly. Please note, due to the color scheme change, the colors used in some other figures (namely, Figs. 1, 6, 7, 9-13, 16-17) are also (slightly) changed. Please check the marked-up version of the manuscript to see these changes in the other figures.

**Revised portions:**

[Figure]

**Figure 14.** Wind speed estimation time series for the NREL-5MW wind turbine. The proposed method ( green line) gives a smooth, less noisy estimate compared to the I&I ( red ) for low (top plots) and high (bottom plots) turbulent cases. The low-frequent component of the wind speed is captured where biased estimation occasionally occurs, potentially due to an inaccurate $C_{\mathrm{p}}$ table utilized in both estimators and an absence of dynamic inflow modeling in the estimators. The actual REWS, as outputted by OpenFAST, is shown  by the  blue line. Zoomed-in plots of 25-s time spans are provided on the right for clearer observation of the

[Figure]

**Figure 15.** Wind speed estimation time series for the IEA-22MW wind turbine. The proposed method gives a smooth, less noisy estimate compared to the I&I ( red ) for low (top plots) and high (bottom plots) turbulent cases. Constant $L_2$ (dashed  orange line) results in lagged estimation compared to gain-tailored $L_2$ (solid  green line). The actual REWS, as outputted by OpenFAST, is shown  by the  blue line. Zoomed-in plots of 25-s time spans are provided on the right for clearer observation of the estimation performance.

Liu, Y., Pamososuryo, A. K., Mulders, S. P., Ferrari, R. M. G., and van Wingerden, J.W.: The Proportional Integral Notch and Coleman Blade Effective Wind Speed Estimators and Their Similarities, 
[revised manuscript text omitted]